# Viral Bcl2s' transmembrane domain interact with host Bcl2 proteins to control cellular apoptosis

Maria Jesús García-Murria[1,3], Gerard Duart[1,3], Brayan Grau[1], Elisabet Diaz-Beneitez[2], Dolores Rodríguez [2], Ismael Mingarro [1] & Luis Martínez-Gil [1✉]

Viral control of programmed cell death relies in part on the expression of viral analogs of the B-cell lymphoma 2 (Bcl2) protein known as viral Bcl2s (vBcl2s). vBcl2s control apoptosis by interacting with host pro- and anti-apoptotic members of the Bcl2 family. Here, we show that the carboxyl-terminal hydrophobic region of herpesviral and poxviral vBcl2s can operate as transmembrane domains (TMDs) and participate in their homo-oligomerization. Additionally, we show that the viral TMDs mediate interactions with cellular pro- and anti-apoptotic Bcl2 TMDs within the membrane. Furthermore, these intra-membrane interactions among viral and cellular proteins are necessary to control cell death upon an apoptotic stimulus. Therefore, their inhibition represents a new potential therapy against viral infections, which are characterized by short- and long-term deregulation of programmed cell death.

[1] Department of Biochemistry and Molecular Biology, Institut de Biotecnologia i Biomedicina, Universitat de València, 46100 Burjassot, Spain. [2] Department of Molecular and Cell Biology, Centro Nacional de Biotecnología, Consejo Superior de Investigaciones Científicas, Campus Universidad Autónoma, 28049 Madrid, Spain. [3]These authors contributed equally: Maria Jesús García-Murria, Gerard Duart. ✉email: Luis.martinez-gil@uv.es

Programmed cell death is indispensable in multicellular organisms, contributing to the balance among cell death, proliferation, and differentiation that is crucial for tissue development and homeostasis[1]. Furthermore, protection from and defense against many disorders, including cancer, autoimmunity, and neurodegeneration, relies on apoptosis and autophagy[2,3]. Pathogen-related diseases are no exception, and efficient control and clearance of infections often require efficient programmed cell death[4,5]. Eliminating infected cells through a controlled death process blocks propagation of the infection and produces danger signals to stimulate an appropriate immune response[6].

Because of its relevance for cell fate, programmed cell death is tightly regulated. One of the primary modulators of apoptosis is the protein family known as B-cell lymphoma 2 (Bcl2)[7]. The Bcl2 family (consisting of ~20 proteins) incorporates pro-survival (e.g., Bcl2 and BclxL)[8], pro-apoptotic (e.g., Bax and Bak[9]), and BH3-only apoptosis activators (e.g., Bid and Bmf)[10]. Proteins are assigned to each subset based on their role in apoptosis (pro- vs anti-apoptotic), as well as on their sequence similarity to Bcl2. Most pro- and anti-apoptotic proteins share all four main Bcl2 sequence homology domains (BH1–4). In contrast, BH3-only members have only the BH3 domain, as their name implies. In addition, many Bcl2 family proteins, including some BH3-only members[11], have a transmembrane domain (TMD) in the carboxyl-terminal (Ct) end that effectively allows for insertion of the protein into the target lipid bilayer[12].

Cellular Bcl2 (cBcl2) proteins can physically interact with each other, forming homo- and hetero-oligomers[9,10,13–16]. These protein–protein interactions (PPIs) constitute an important regulatory mechanism of programmed cell death. In healthy cells, anti-apoptotic Bcl2s inhibit the activation of pro-apoptotic proteins, either through direct interaction or sequestering BH3-only activators[7]. Upon an apoptotic stimulus, BH3-only and pro-apoptotic proteins are discharged, promoting mitochondrial membrane permeabilization and the release of cytochrome c into the cytosol which, in turn, will activate the apoptosome. Interactions among Bcl2 family members have been thought to occur only through soluble domains, especially BH domains. However, recent findings suggested that Bax TMD interacts with pro-survival Bcl2 proteins[17], expanding the range of interactions involved in the control of programmed cell death.

Viruses have developed multiple strategies to modulate cell death, including masking of internal cellular sensors, caspase regulation, signaling cascade modulation, and mimicking of Bcl2 regulators[18,19]. Functional homologs of cBcl2, known as viral Bcl2s (vBcl2s; singular, vBcl2), are present in numerous viral families, including *Herpesviridae*, *Poxviridae*, *Adenoviridae*, and *Birnaviridae*[20]. The sequence homology between vBcl2 and cBcl2 proteins varies considerably. However, the crystal structures of some vBcl2s reveal a structural homology in key domains between cBcl2 and vBcl2 despite no sequence homology, suggesting functional commonalities[5,21].

Here, we show that the Ct hydrophobic regions of herpesviral and poxviral vBcl2s can operate as TMDs, effectively inserting the protein into its target membrane. The vBcl2 TMDs facilitate homo-oligomerization and hetero-oligomerization with pro- and anti-apoptotic cBcl2 TMDs inside the membrane and in the absence of apoptotic stimuli. These newly discovered interactions participate in the regulation of cBcl2, thus modulating programmed cell death upon apoptotic stimuli. Furthermore, our results suggest that inhibitors of these intramembrane PPIs could be used therapeutically against life-threatening viral infections characterized by short- and long-term deregulation of apoptosis.

## Results

**vBcl2 contain a functional Ct TMD.** Like their cellular counterparts, many vBcl2s present a hydrophobic region in their Ct end. To identify whether these hydrophobic regions could act as true TMDs, we selected six vBcl2 proteins from two distinct viral families (3 herpesviruses and 3 poxviruses): BHRF1 (*Human gammaherpesvirus 4 – Epstein–Barr virus*, HHV4)[22], ORF16 (*Human gammaherpesvirus 8 – Kaposi's sarcoma–associated herpesvirus*, HHV8)[23], ORF16 (*Bovine gammaherpesvirus 4*, BoHV4)[24], F1L (*Vaccinia virus*, VacV)[25,26], M11L (*Myxoma virus*, MyxV)[26,27], and ORFV125 (*Orf virus*, OrfV)[28]. To avoid confusion, here we use the viral acronym to refer to the vBcl2 protein.

First, we analyzed in silico the presence of TMDs in the selected vBcl2. For this purpose, we used two TMD prediction software packages, TMHMM v2.0[29,30] and the ΔG prediction server[31,32]. Both algorithms identified a TMD in the Ct of HHV4, HHV8, MyxV, and OrfV. Although the ΔG prediction server identified a Ct TMD for BoHV4 and VacV, the TMHMM software did not (Supplementary Fig. 1).

We then aimed to explore the membrane insertion capacity of the predicted membrane-spanning segments using an in vitro assay based on the *E. coli* leader peptidase (Lep), an assay that allows for accurate and quantitative description of the membrane insertion capability of short sequences. The Lep sequence we employed contains an extended Nt luminal section, followed by two TMDs (H1 and H2) connected by a cytoplasmic loop, and a large luminal Ct domain. Two glycosylation acceptor sites were inserted into the Lep (G1 and G2), one in each luminal domain (Nt and Ct ends) (Fig. 1a). N-linked glycosylation occurs exclusively in the lumen of the endoplasmic reticulum (ER) because of the location of the oligosaccharyltransferase (OST) active site (a translocon-associated enzyme responsible for the oligosaccharide transfer[33]), thus serving as a topological marker. Glycosylation of an acceptor site increases the apparent molecular mass of the protein (~2.5 kDa), which facilitates its identification by gel electrophoresis. When H1 and H2 are present, both G1 and G2 acceptor sites locate in the ER lumen and are subsequently glycosylated. Likewise, if H2 is replaced by a sequence that the translocon efficiently recognizes as a TMD, a double (G1 and G2) glycosylation would be expected. In contrast, if H2 is deleted or substituted by a protein segment that the translocon does not recognized as a TMD, only G1 would remain in the lumen, and single glycosylation will be observed (Fig. 1a).

Using this system, we studied the insertion of all six vBcl2 hydrophobic Ct domains (Fig. 1b). Additionally, the single TMD of two mitochondrial membrane proteins, Tomm20 and Tomm22 (T20 and T22), together with a construct in which Lep was not altered were used as insertion controls. As a mono-glycosylated control, we used the Ct region of the BH3-only protein Noxa, a hydrophobic domain not recognized by the translocon as a TMD[11]. The sequences for all non-viral proteins referenced here can be found in Supplementary Fig. 2. Additionally, we used a Lep mutant from which H2 was deleted as a mono-glycosylated/non-insertion control (Supplementary Fig. 3). To ensure that the increase in molecular weight was due to glycosylation of the acceptor sites, OST samples were incubated in the presence (+) or absence (−) of Endoglycosydase H (EndoH), a glycan-removing enzyme. The results of the glycosylation assay indicated that all vBcl2 hydrophobic regions, including those derived from BoHV and VacV, were efficiently inserted into ER-derived microsomal membranes (Fig. 1c). Furthermore, the Lep assay revealed that all viral TMDs insert into ER-derived membranes more efficiently than predicted by the ΔG prediction server (Fig. 1b).

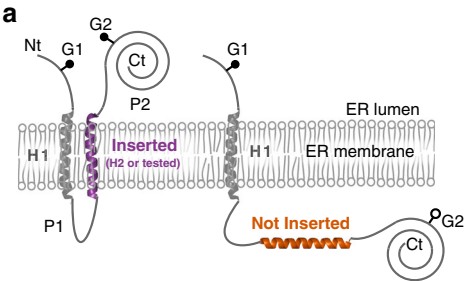

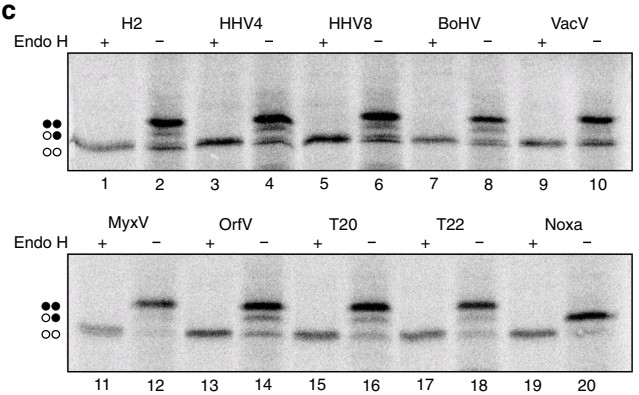

**Fig. 1 Insertion of vBcl2 Ct hydrophobic sequences in eukaryotic membranes. a** A schematic representation of the leader peptidase (Lep) model protein. G1 and G2 denote artificial glycosylation acceptor sites. The sequence under investigation was introduced replacing the H2 domain of Lep. Recognition of the tested sequence as a TMD by the translocon machinery (highlighted in purple) results in the modification of the G1 and G2 acceptor sites. The Lep chimera will be singly glycosylated if the sequence being tested is not recognized as a TMD and thus not inserted into ER-derived membranes (shown in orange). **b** The sequences (including name and organism of origin) used in the assay are shown alongside their predicted $\Delta G$ ($\Delta G_{pred}$) in kcal/mol, calculated using the $\Delta G$ prediction server in its $\Delta G$ prediction mode. Residues shown in gray were not considered by the $\Delta G$ prediction server. Additionally, the experimental $\Delta G$s in kcal/mol ($\Delta G_{exp}$), calculated based on the ratios between mono and double glycosylated populations, are presented in the right column. Experimental results are the average of at least four independent experiments. Purple numbers indicate negative $\Delta G$s (insertion of the tested sequence), while orange numbers denote $\Delta G$ values above 0. **c** A representative example ($n = 4$) of an in vitro protein translation in the presence of ER-derived microsomes and in the presence ($+$) or absence ($-$) of Endoglycosydase H (EndoH), a glycan-removing enzyme. The absence of glycosylation of G1 and G2 acceptor sites is indicated by two white dots, single glycosylation by one white and one black dot, and double glycosylation by two black dots.

**Homo-oligomerization of the vBcl2 TMDs**. cBcl2 TMD homo-oligomerization has been reported in biological membranes[17]. To assess whether vBcl2 TMDs exhibit this self-associating property, we employed bimolecular fluorescent complementation (BiFC) approach[34] adapted for the study of intramembrane interactions[17,35]. Briefly, TMDs were fused with two non-fluorescent fragments of the venus fluorescent protein (VFP), the VN (N terminus) and the VC (C terminus; Fig. 2a). Interaction of the TMDs facilitated the approximation of the VN and VC ends and the reconstitution of the VFP protein structure and, in consequence, the recovery of its fluorescent properties. As a positive control and normalization tool, we included the TMD of glycophorin A (GpA), a sialoglycoprotein found in human erythrocyte membranes that can form non-covalent dimers solely through the association of its single hydrophobic TMD[36–38]. The non-oligomerizing TMDs of the mitochondrial membrane proteins T20 and T22 and H2 from Lep were used as negative controls for membrane overcrowding (Fig. 2b).

The TMDs of HHV4, HHV8, VacV, MyxV, and OrfV showed an interaction capability above the controls, including T20 (used for the statistical analysis) and similar to that observed with Bcl2 TMD (Fig. 2b). However, BoHV did not show VFP fluorescence values significantly higher than the T20 control (Fig. 2b). Western blot analysis showed comparable expression levels of all chimeras (Supplementary Fig. 4). Because of the large variability, we decided to revise the results obtained with BoHV TMDs. For this purpose, we used BlaTM, a genetic tool designed to study TMD–TMD interactions in bacterial membrane[39].

Briefly, to measure interactions between TMDs, the designed sequences were fused to either the Nt or the Ct end of a split β-lactamase (βN and βC, respectively) and to the green fluorescent protein (GFP) (Supplementary Fig. 5a). These chimeras also included the pelB cleavable signal peptide, which directs the protein to the inner bacterial membrane and determines its topology, ensuring a periplasmic localization of the β-lactamase. Once located in the inner membrane, an efficient TMD–TMD interaction facilitates the reconstitution of the β-lactamase and thus the growth of the bacteria in selective media (ampicillin). In this assay, the LD50 of the antibiotic served as an indicator of the interaction strength (Fig. 2c, d), and GFP fluorescence allowed for rapid quantification of protein levels (Supplementary Fig. 5b). As in the previous experiments, the TMD of GpA was used as a positive control and normalization value across experimental replicas, and the TMD of T20 was used as a negative control. Additionally, the HHV8 TMD was included in the assay as an extra control. In the BlaTM assay, both BoHV TMD and HHV8 TMD showed an interaction (ampicillin LD50) above the negative control (Fig. 2c, d). Expression levels (measured using the GFP fluorescence) were comparable for GpA, T20, HHV8, and BoHV (Supplementary Fig. 5b).

By directly localizing to the mitochondria, Bcl2 controls mitochondrial outer membrane (MOM) permeabilization and, subsequently, apoptosis. In the mitochondria, Bcl2 can interact through its TMD with other members of the Bcl2 family[17]. Nonetheless, Bcl2 also localizes to other intracellular compartments, including the ER and the Golgi apparatus[40]. To analyze the location of the observed vBcl2 TMD homo-oligomers, we expressed the appropriated BiFC partner combinations together with an ER, mitochondrial, or plasma membrane marker (Supplementary Fig. 6). Our results showed that all the assayed TMD homo-oligomers occurred at the ER and at the mitochondrial membranes. Of interest, the TMD oligomers of VacV and MyxV could also be observed in the plasma membrane. These results suggest a similar distribution for vBcl2 and cBcl2.

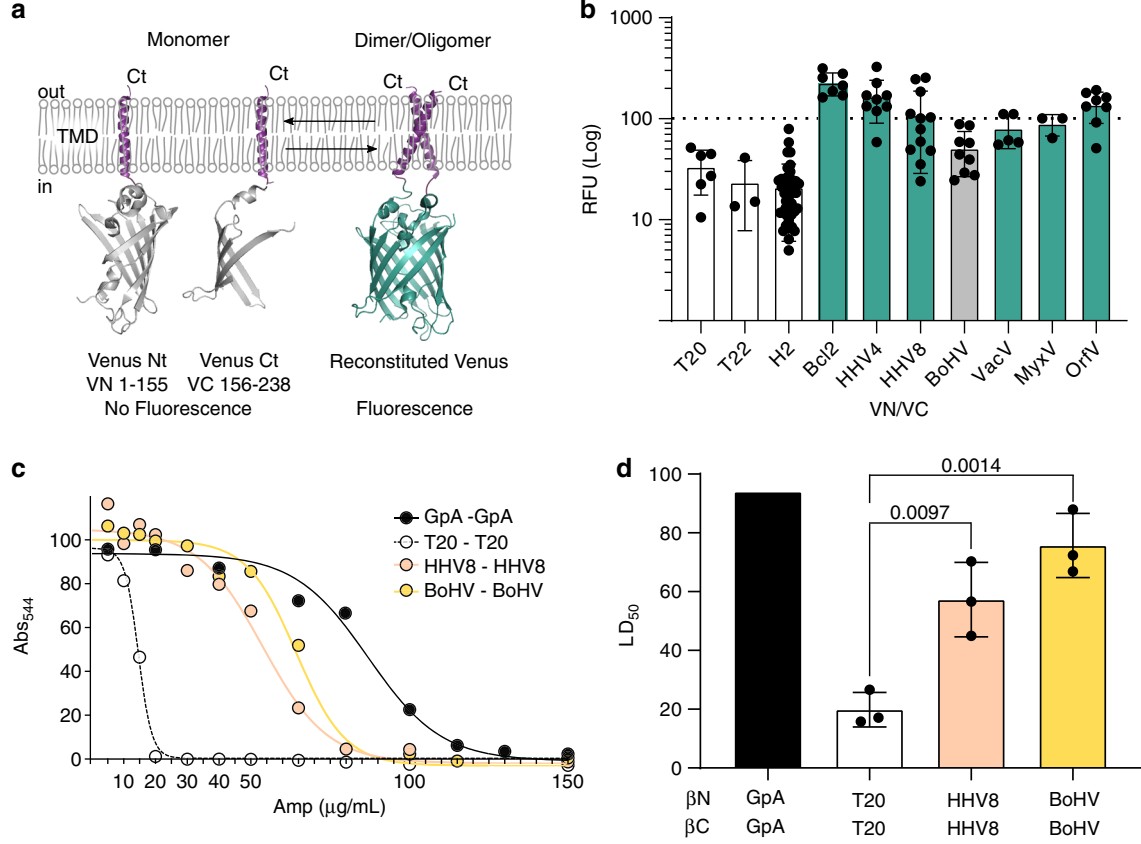

**Fig. 2 Homo-oligomerization in biological membranes. a** A schematic representation of the Bimolecular Fluorescent Complementation (BiFC) assay. The position of the Nt and Ct ends and the TMDs included in the constructs are indicated. The residues included in each VFP fragment are indicated below the protein representation. **b** Relative fluorescence units (RFU) of each tested homo-oligomer in the BiFC assay. The mean and standard deviation of at least three independent experiments are represented ($n$ values 6, 3, 37, 9, 9, 12, 9, 5, 3, 9, respectively). The individual value of each experiment is represented by a solid dot. The VN-GpA/VC-GpA homodimer, used as a positive control and to normalize values across experiments, is represented by a dotted line. The T20, T22, and Lep H2 TMDs were included as negative controls (white bars). Those homo-oligomer that produced fluorescence levels significantly higher than the T20 TMD homo-oligomer (two-tailed homoscedastic $t$-test) are highlighted in green, and those that did not are shown in light gray. **c** Representative examples of dose–response curves used to calculate $LD_{50}$ values in the BlaTM assay. The TMD homodimer of GpA was used as a positive control (black), while the T20 TMD was used as a negative control (white). **d** BlaTM calculated mean and standard deviation $LD_{50}$ of at least three independent experiments ($n = 3$). The individual value of each experiment is represented by a solid dot. The βN-GpA/βC-GpA homodimer was used as a positive control (black bar). The level of significance ($p$-value, two-tailed homoscedastic $t$-test) when comparing T20 (white bar) vs HHV8 (pink bar) or BoHV (yellow bar) homo-oligomers is shown above the bars.

**cBcl2 and vBcl2 TMDs hetero-oligomerize in eukaryotic membranes**. Next, we decided to investigate the potential TMD–TMD interactions between vBcl2s and cBcl2s. For this purpose, we used the previously described BiFC approach. For any given interaction (i.e., $X/Y$), two possible combinations with the VFP fragments exist (VN-$X$/VC-$Y$ and VN-$Y$/VC-$X$), so we investigated both. Once again, the GpA TMD homodimer was used as a positive control and a normalization value. As negative controls, we used the interaction of each partner in the hetero-oligomers with T20 TMD (i.e., for the $X/Y$ interaction, $X$/T20 and T20/$Y$). An interaction was considered only if the fluorescence values were significantly higher than those of the two particular negative controls. To reinforce our data, we included in the study the interactions between the vBcl2 TMDs and T22 or H2 TMDs. The mean relative fluorescence and standard deviation of all the controls and the interactions between vBcl2 TMD and T22 TMD and Lep H2 can be found in Supplementary Fig. 7. The assay was carried out in human-derived cells despite the natural host of the virus; however, the amino acid composition of the cBcl2s TMDs are well conserved across the corresponding species (Supplementary Fig. 8). First, we investigated the potential TMD–TMD

interactions between vBcl2s and the anti-apoptotic cBcl2s Bcl2 and BclXL (Fig. 3a, b). All of the viral TMDs could interact with the TMD of Bcl2 in at least one of the assayed combinations. On the other hand, although the majority of vBcl2 TMDs also interacted with BclXL TMDs, BoHV and MyxV TMDs did not (regardless of the partner combination used in the screening).

Next, because of the strength and consistency of the observed signal, we decided to further characterize the Bcl2-HHV8 TMD–TMD interaction. First, we corroborated this intramembrane interaction by performing a competition assay. The Bcl2 and HHV8 TMD-driven homo-oligomers were challenged either with the Bcl2 full-length protein (Bcl2-FL) or with a Bcl2 mutant lacking the TMD (Bcl2 ΔTMD). The Bcl2 TMD homo-oligomer was hindered when Bcl2-FL was co-expressed but not when the Bcl2 ΔTMD construct was included in the assay (Fig. 4a). Similarly, the addition of Bcl2-FL to the HHV8 TMD homo-oligomer decreased the observed fluorescence compared with the effect of the presence of Bcl2 ΔTMD. This result confirmed that HHV8 and Bcl2 TMDs can interact and suggested that the soluble domain of Bcl2 does not preclude an interaction between TMDs.

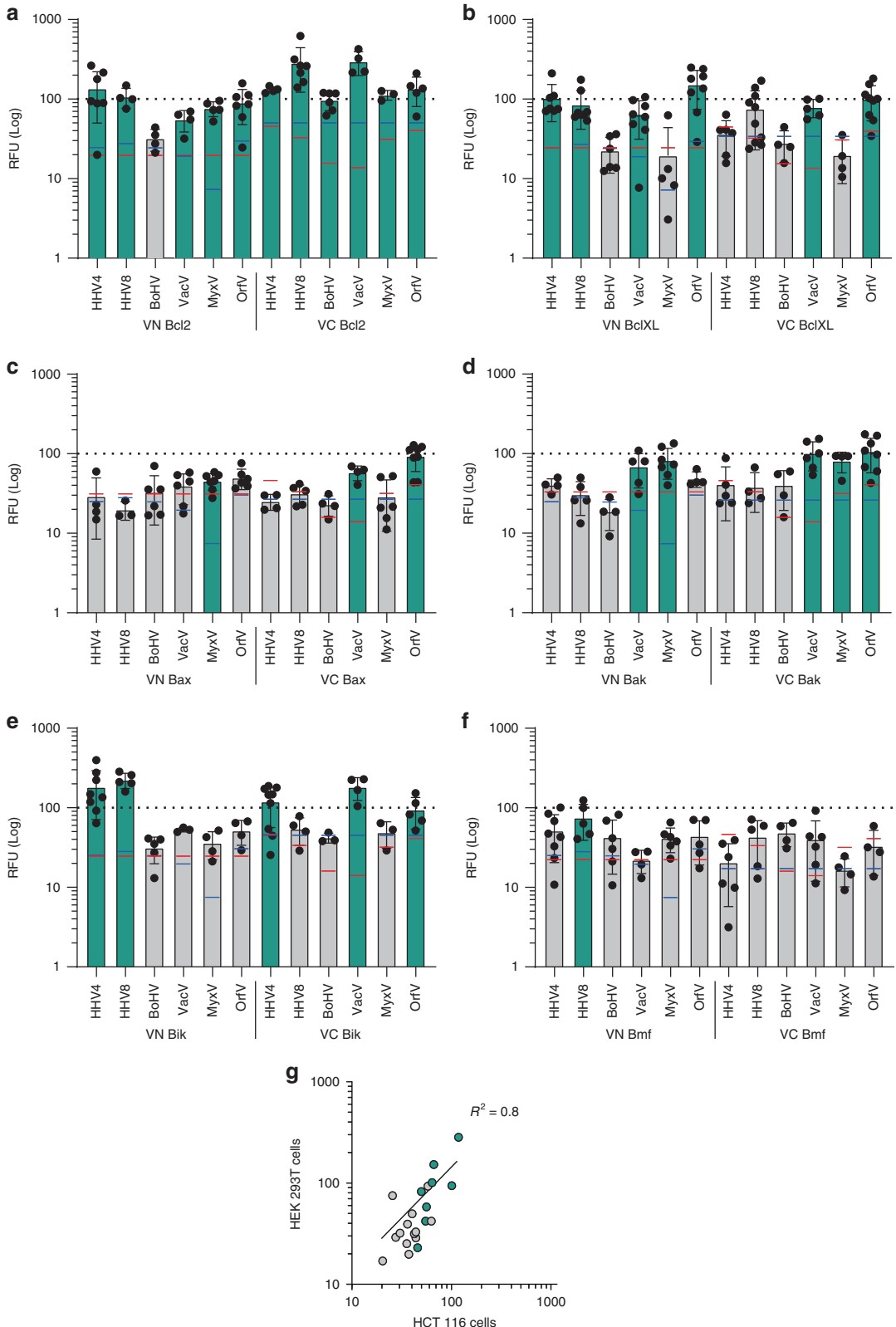

**Fig. 3 Hetero-oligomerization in Eukaryotic membranes. a–f** Relative fluorescence (RFU) for the hetero-oligomerization of vBcl2 and cBcl2 TMDs. The mean and standard deviation of at least three independent experiments are shown. Solid dots represent the results of individual experiments (*n* ranges between 3 and 10). The TMD included in the VFP chimeras (VN or VC) is indicated below each bar. The GpA TMD homodimer was used as a positive control and as the normalization value (dotted line). The interactions of each partner in the hetero-oligomers with T20 TMD were used as negative controls (i.e., *X*/T20 and T20/*Y* for the *X*/*Y* interaction). The blue (VN/T20 TMD) and red (VC/T20 TMD) lines within each bar indicate the fluorescence of the controls. An interaction (highlighted in green) was considered only if the RFU obtained was significantly higher (two-tailed homoscedastic *t*-test, *p*-value < 0.05) than those of both negative controls. **g** Correlation of the BiFC assay results in HCT 116 and HEK 293T cells. Green dots indicate interactions in HEK 293T cells. A linear trend line and the corresponding coefficient of correlation are shown.

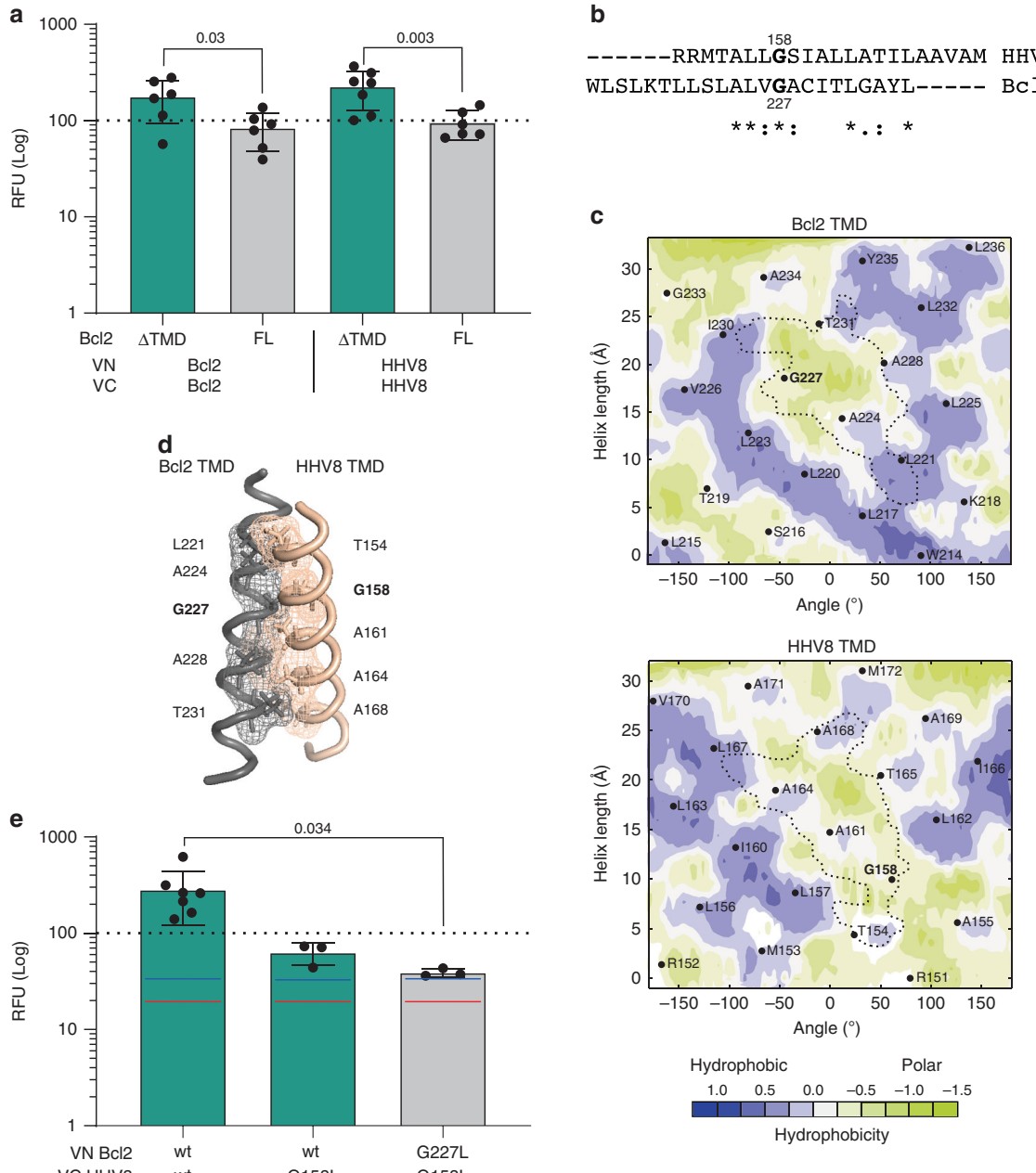

**Fig. 4 Interaction between HHV8 and Bcl2 TMDs. a** The BiFC-derived fluorescence (RFU) for the homo-oligomerization of Bcl2 (left) or HHV8 (right) TMDs was measured in the presence of Bcl2 (FL) or Bcl2 ΔTMD (ΔTMD) in HEK 293T cells. The mean and standard deviation of at least six independent experiments are shown ($n \geq 6$). Solid dots represent the results of individual experiments. The GpA TMD homodimer was used as a positive control and as the normalization value (dotted line). A significant (two-tailed homoscedastic $t$-test) decrease in the fluorescence is highlighted in gray, $p$-values are indicated. **b** Sequence alignment of Bcl2 and HHV8 TMDs. The alignment was done with Clustal Omega (EMBL-EBI). **c** Predicted (PredDIMER) contact area between Bcl2 and HHV8 TMDs. The position (length and angle) of each residue in a putative α-helix are indicated. A dotted line encircles the contact area in each TMD. The hydrophobicity of the residues is shown with a color scale. As expected, there are many hydrophobic residues in both TMDs. However, the contact area between both α-helices is mostly hydrophilic. **d** Model of a putative dimer between Bcl2 TMD and HHV8 TMD, obtained with PredDIMER. The residues involved in the interaction are indicated, conserved glycine residues in Bcl2 and HHV8 are highlighted in bold. **e** Bcl2 and HHV8 TMD hetero-oligomerization measured by BiFC in HEK 293T cells. Fluorescence from the VN GpA/VC-GpA homodimer was used to normalize the signal (dotted line). The mean and standard deviation of at least three independent experiments are shown ($n \geq 3$). Solid dots represent the results of individual experiments. The blue (VN T20 TMD) and red (VC T20 TMD) lines indicate the fluorescence of the controls used for the statistical analysis (two-tailed homoscedastic $t$-test), green bars denote RFU values above the controls, $p$-values are indicated.

An in silico analysis of the potential dimer interface (done with Preddimer, a TM segment dimer prediction software program[41]) suggested that conserved glycine residues between Bcl2 and HHV8 TMDs (at positions 158 and 227, respectively) could be located at the contact surface of a putative HHV8-Bcl2 TMD–TMD heterodimer (Fig. 4b–d, and Supplementary Fig. 9a). This positioning would create the appropriate environment for ridge–groove arrangements, as observed for other interacting TMDs[36,37,42]. Single amino acid substitutions were designed to analyze the role of these glycine residues. Precisely, a mutation of

glycine 158 to leucine in HHV8 TMD (G158L) reduced the observed interaction between Bcl2 and HHV8 TMDs in the BiFC assay (Fig. 4e). Moreover, mutation of glycine residues in both helices (G158L in HHV8 and G227L in Bcl2) further decreased the fluorescence signal to levels significantly below those obtained with the wild-type forms of the TMDs. Of note, the introduction of G158L in HHV8 or G227L in Bcl2 did not alter insertion into biological membranes (Supplementary Fig. 3). These results indicate that the TMDs of Bcl2 and HHV8 can interact efficiently and specifically at the mitochondrial and ER membranes through arrangements in which the glycine residues have a crucial role.

We next analyzed the interactions between vBcl2 TMDs and the TMDs of the cellular pro-apoptotic Bax and Bak proteins (Fig. 3c, d). All three poxviral TMDs (VacV, MyxV, and OrfV) could interact with Bax and Bak TMDs in the absence of apoptotic stimuli and independent of any cytosolic (soluble) domain interactions. On the other hand, herpesviral HHV4, HHV8, and BoHV TMDs showed no interaction with either Bax or Bak TMDs. MyxV protein interacts with Bak and Bax[43,44] inhibiting the conformational activation of Bax[43]. Based on this information, we decided to investigate the role of MyxV TMD and its interaction with Bax TMD in the viral protein function. For this purpose, we first decided to confirm the host–pathogen intramembrane interaction using the previously described BlaTM assay. The MyxV and Bax TMDs were inserted into the βN and βC chimeras, respectively, and assayed together or in combination with the complementary chimeras carrying the TMD of T20 (Fig. 5a, b). The results of the BlaTM assay confirmed the interaction between MyxV and Bax TMDs, which showed $LD_{50}$ values that were significantly higher than the βN-T20/βC-T20, βN-MyxV/βC-T20, and βN-T20/βC-Bax combinations.

Using the Preddimer algorithm, we analyzed the Bax TMD–MyxV TMD interaction (Fig. 5c, d and Supplementary Fig. 9b). Once again, the model showed an interaction in which a ridge–groove arrangement was created by an adequate disposition of large and small residues. Of interest, a large proportion of aromatic residues was found at the interface of these two TMDs. Amino acid substitutions were again designed to analyze the role of the small residues. Mutation of glycine 158 to isoleucine in MyxV TMD (G158I) or double mutation of glycine 179 and alanine 183 to isoleucine in Bax TMD (G179I A183I) reduced the interaction of Bax and MyxV TMDs in the BlaTM assay (Fig. 5e). Moreover, the introduction of mutations in both helices (G158I in MyxV and G179I A183I in Bax) further decreased antibiotic resistance. Of note, mutations G158I in MyxV and G179I A183I in Bax TMD did not alter membrane insertion (Supplementary Fig. 3).

In eukaryotic cells, ectopic expression of Bax TMD induces some caspase 3/7 activation and subsequently cell death, which expression of the anti-apoptotic Bcl2 TMD can counteract, thanks to the interaction among Bcl2 and Bax TMDs[17]. We wondered whether the interaction between MyxV and Bax TMDs also could preclude Bax TMD caspase activation. To test this hypothesis, we co-expressed H2, Bcl2, or MyxV TMD with Bax TMD and measured the resulting cell viability (Fig. 5f). In agreement with the hetero-oligomerization results, Bcl2 and MyxV TMDs, but not control H2 TMD, interfered with the Bax TMD-induced apoptosis.

Next, to extend the BiFC-based screening of the intramembrane host–pathogen interactions between vBcl2s and cBcl2s, we studied the interactions with the BH3-only apoptotic modulators Bik and Bmf (Fig. 3e, f). The TMDs of HHV4, HHV8, VacV, and OrfV could interact with the Bik TMD. However, according to our results, viral interactions with Bmf were much sparser, and only the HHV8 TMD could interact with Bmf TMD. All intramembrane PPIs found between cellular and viral proteins are

summarized in Supplementary Fig. 10. For a more comprehensive visualization, we also have included a network representation (Supplementary Fig. 11). Additionally, the result of the BiFC-based HEK 293T screening was confirmed in HCT 116 cells. Despite some differences, a strong correlation between the BiFC assay in both cell types can be observed ($R^2 = 0.8$) (Fig. 3g and Supplementary Fig. 12).

While studying interaction with the BiFC assay (i.e., $X/Y$), we observed some differences in the results obtained with each of the two possible combinations (VN-$X$/VC-$Y$ and VN-$Y$/VC-$X$).

Due to the nature of the BiFC assay, it is possible, as has occurred in the case of the BlaTM assay, that the reporter signal resulting from a TMD–TMD interaction depends not only on the sequence of the interacting TMDs, and thus its inherent affinity, but also on the orientation of the interacting surfaces of the TMDs in relation to the accompanying signaling domains[39]. To test our hypothesis we chose the first of the interactions where significant differences between the two possible BiFC combinations were found, i.e., the interaction between Bcl2 and BoHV TMDs (Fig. 3a). Our screening revealed VN-BoHV/VC-Bcl2 as a TMD–TMD interaction but not VN-Bcl2/VC-BoHV. To change the orientation of the TMD with respect to the reporter domain we inserted one (+1), two (+2), or three (+3) native residues at the Nt end of BoHV TMD[39]. The observed fluorescence for the interaction of the new BoHV TMD variants in combination with the Bcl2 TMD were higher (and significantly above their controls) than the interaction with the original design of the VN-BoHV chimera (Supplementary Fig. 13). Although this result confirms our hypothesis, due to a large number of interactions studied and despite a large number of replicas, we cannot rule out the presence of some false positive or negative results.

Collectively, these results point to an intricate network of interactions between the TMDs of viral and cellular origin. A comprehensive analysis of the data using a principal component analysis (PCA) revealed that HHV4 and HHV8 TMDs from herpesviral proteins behave similarly, as would have been expected given their taxonomic proximity (Fig. 6a, b). Likewise, the TMDs of a poxviral origin, particularly VacV and OrfV TMDs, were tightly grouped in the PCA. However, the observed similarities could not have been inferred by the analysis of the TMDs sequences (Fig. 6c), which suggests a structural pattern underlying the sequence that governs the TMD interactions.

**Viral–host TMD–TMD interactions are necessary to modulate apoptosis.** To identify whether these newly found viral–host TMD–TMD interactions are necessary to control cellular apoptosis, we transfected HeLa cells with Bcl2, HHV8, or MyxV either with or without the TMD (FL and ΔTMD, respectively). Additionally, we included chimeras in which the TMD of each of the previously mentioned proteins was replaced by the TMD of T20 (Bcl2-T20, HHV8-T20, and MyxV-T20, respectively), which our previous results indicated cannot interact with any cBcl2 TMD. Once transfected, cells were treated with doxorubicin to induce apoptosis[45].

As expected, the FL proteins could prevent apoptosis (Fig. 7a, and Supplementary Fig. 14). However, once the TMD was removed, none of the proteins could promote survival or stop apoptosis. Similarly, the chimeras carrying the TMD of T20 could not control the doxorubicin-induced apoptosis, suggesting that the function of the viral TMDs extends beyond anchoring the protein to the membrane and that TMD–TMD hetero-oligomerizations are crucial for modulating viral-induced cell death (Fig. 7a, b). Western blot analysis confirmed that differences in expression levels were not the source of the observed results (Fig. 7c).

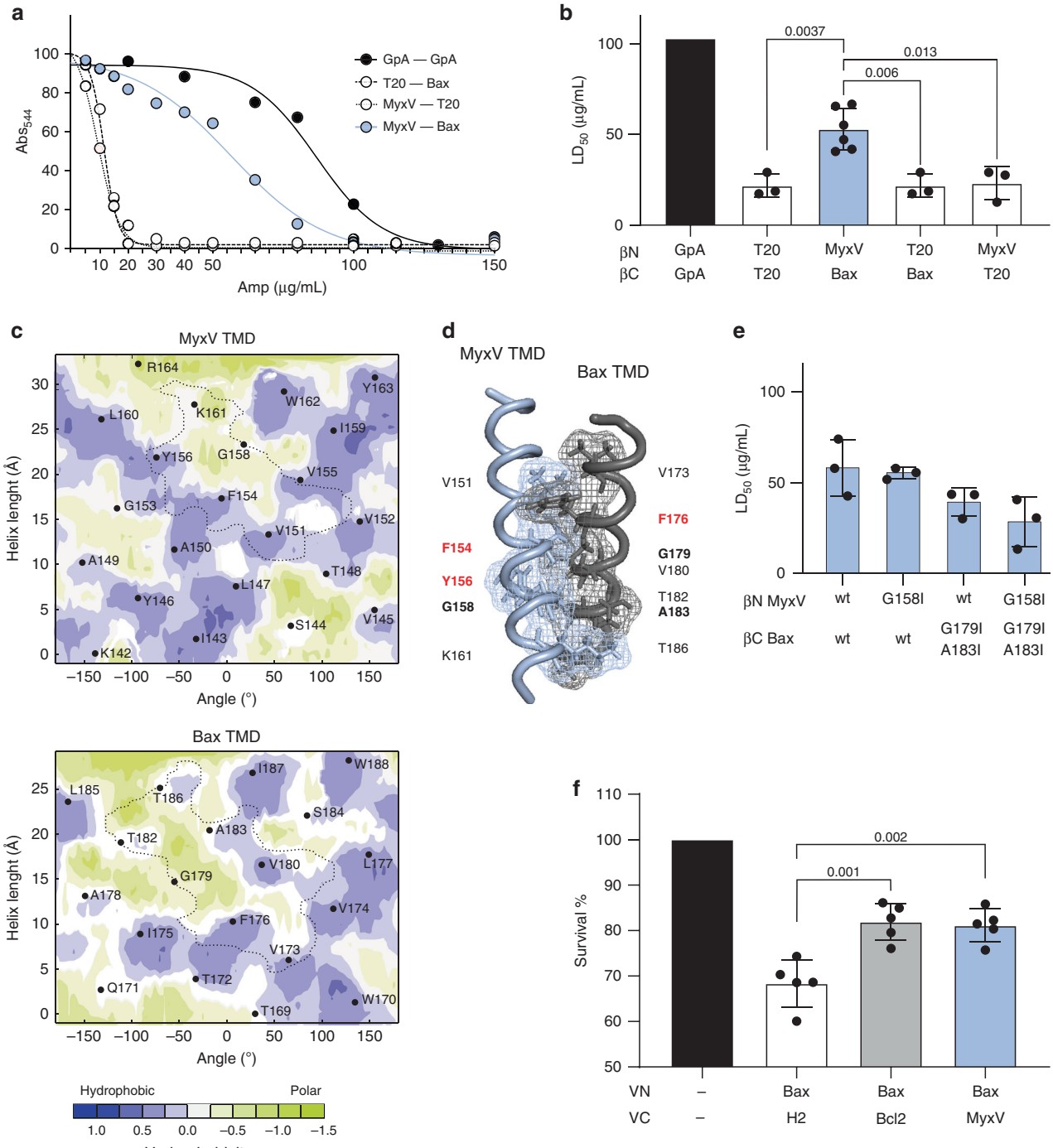

**Fig. 5 Interaction between MyxV and Bax TMDs. a** Representative example of dose–response curves used to calculate LD$_{50}$ values in the BlaTM assay. The TMD homodimer of GpA was used as a positive control (black), while the T20 TMD was used as a negative control (white). **b** Summary of the results for the BlaTM assay. The indicated βN and βC chimeras were expressed in *E. coli* and the resulting ampicillin LD$_{50}$ measured. LD$_{50}$ mean and standard deviation of at least three independent experiments ($n \geq 3$) are shown. Two-tailed homoscedastic *t*-test *p*-values are indicated. Solid dots represent the results of individual experiments. **c** Predicted (PredDIMER) contact area between MyxV and Bax TMDs. The position (length and angle) of each residue in a putative α-helix is indicated. A dotted line encircles the contact area in each TMD. The hydrophobicity of the residues is shown with a color scale. **d** Model of a putative dimer between MyxV TMD and Bax TMD, obtained with PredDIMER. The residues involved in the interaction are indicated. Glycine and alanine residues are highlighted in bold while aromatic residues are shown in red. **e** BlaTM analysis for the interaction between MyxV and Bax TMDs carrying substitutions of Gly 158 (MyxV) and/or Gly 179 and Ala 183 (Bax) by Ile. LD$_{50}$ mean and standard deviation of at least three independent experiments are shown ($n \geq 3$), solid dots represent the results of individual experiments. **f** Bax TMD-induced cell death. Cells were transfected with Bax TMD and H2, Bcl2 or MyxV TMD or mock-transfected; 24 hpt cell viability was measured by Trypan blue staining. Survival percentage mean and standard deviation of five independent experiments ($n = 5$) are shown. Two-tailed homoscedastic *t*-test *p*-values are indicated. Solid dots represent the results of individual experiments.

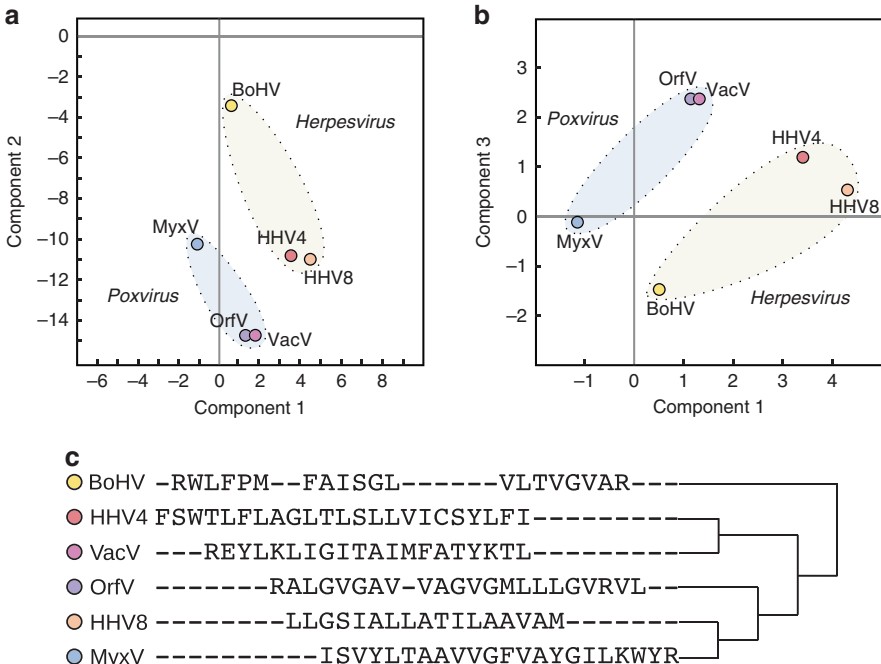

**Fig. 6 TMD interaction and sequence profile. a**, **b** Principal component analysis (PCA) of vBcl2 TMDs interaction profile with host TMDs. The colored areas highlight the viral families, with poxviruses in light blue and herpesviruses in light yellow. The analysis was performed using only the BiFC assay results. **c** vBcl2 TMDs Multiple Sequence Alignment. Results obtained with Clustal Omega (EMBL-EBI) using the default parameters.

Despite T20 being a mitochondrial-localized protein, the substitution of the natural TMD on Bcl2, HHV8, or MyxV by the T20 TMD could affect the localization of these proteins and impede their anti-apoptotic role. To investigate this possibility, we compared the localization of all three protein variants (FL, ΔTMD, and T20) by fluorescent confocal microscopy. The confocal analysis revealed that substitution of the natural TMD with the T20 TMD did not alter the localization of the protein (Supplementary Fig. 15). Furthermore, colocalization with a mitochondrial marker was observed with the Bcl2, HHV8, and MyxV-FL and all T20 variants. In contrast, the elimination of the Ct TMD generated soluble proteins in all three cases. These results suggest that the TMD, present in HHV8 and MyxV, not only facilitated proper localization of the proteins but is also necessary to establish interactions with host TMDs, which are essential for the viral control of apoptosis.

Next, we transfected HeLa cells with Bcl2 and MyxV (FL and T20 variants for both proteins) and induced apoptosis with Bax-FL (Fig. 7d). Both Bcl2 and MyxV-FL proteins could rescue Bax-induced apoptosis. However, the substitution of the TMD in either Bcl2 or MyxV by the Ct membrane anchoring segment of T20 eliminated their anti-apoptotic properties. Collectively, these results not only suggest a potential role for the Bax–MyxV TMD–TMD interaction during the viral infection but also reinforce our screening results.

It has been proposed that MyxV helices 2, 3, 4, and 5 form a binding groove for Bak and Bax[46]. Mutations in key positions of this groove alter MyxV binding to the aforementioned pro-apoptotic cellular proteins and subsequently disrupt its anti-apoptotic action. To compare the influence of in-groove vs TMD interactions, HeLa cells were transfected with MyxV-FL, T20, or MyxV bearing substitutions in alanines 71 or 82 to phenylalanines (A71F and A82F respectively)[46]. To induce apoptosis, cells were co-transfected with Bax-FL. Our results confirmed that mutations in the binding groove of MyxV, primarily A82F, alter MyxV pro-survival function (Supplementary Fig. 16a). However, the substitution of MyxV TMD by T20 TMD produced a stronger

effect on the ability of the protein to rescue Bax-induced apoptosis. Western blot analysis confirmed comparable expression levels for all of the FL, T20 A71F, and A82F variants (Supplementary Fig. 16b).

**vBcl2 and cBcl2 TMDs interactions control viral-induced apoptosis.** Finally, we aimed to explore whether vBcl2 TMD interactions are necessary to control the apoptosis induced by a viral infection. For this purpose, we infected HeLa cells with VacV-PKR, a VacV recombinant strain expressing the double-stranded RNA-dependent protein kinase (PKR) and capable of inducing apoptosis[47] and transfected with plasmids carrying Bcl2, Bcl2-ΔTMD, Bcl2-T20, HHV8, HHV8-T20, MyxV, or MyxV-T20 (under the control of an IPTG inducible promoter) or mock-transfected with an empty plasmid. At 24 h.p.i., levels of caspases 3 and 7 activity and cell death were measured as a readout for apoptosis (Fig. 8a, b). Bcl2-FL protein could partially block the PKR-induced apoptosis[48]. However, Bcl2-ΔTMD or Bcl2-T20 did not rescue cell death. Similarly, HHV8-FL and MyxV-FL significantly reduced caspase 3/7 levels and decreased cell death. On the other hand, HHV8-T20 or MyxV-T20, which includes a TMD incapable of establishing intramembrane interactions, did not reduce caspase 3/7 levels and were unable to control cell death. To confirm these results, we infected HeLa cells with a Modified Vaccinia Ankara virus strain lacking F1L protein (MVA-ΔF1L) and capable of inducing apoptosis[49] and transfected with Bcl2-FL, Bcl2-ΔTMD, Bcl2-T20, HHV8-FL, HHV8-T20, MyxV-FL, MyxV-T20, VacV-FL, pr VacV-T20 (under the control of IPTG) or mock-transfected with an empty plasmid (Fig. 8c, d). Additionally, the use of MVA-ΔF1L allowed us to explore the role of TMD interactions in VacV vBcl2 (F1L). As in the previous experiment (VacV-PKR, panels a and b), only proteins bearing a TMD capable of establishing interactions (FL variants) were able to control cellular apoptosis. In both experiments, the western blot analysis confirmed comparable expression levels for all the FL, ΔTM, and T20 constructs (Fig. 8e). In summary, our results demonstrate that interactions between viral

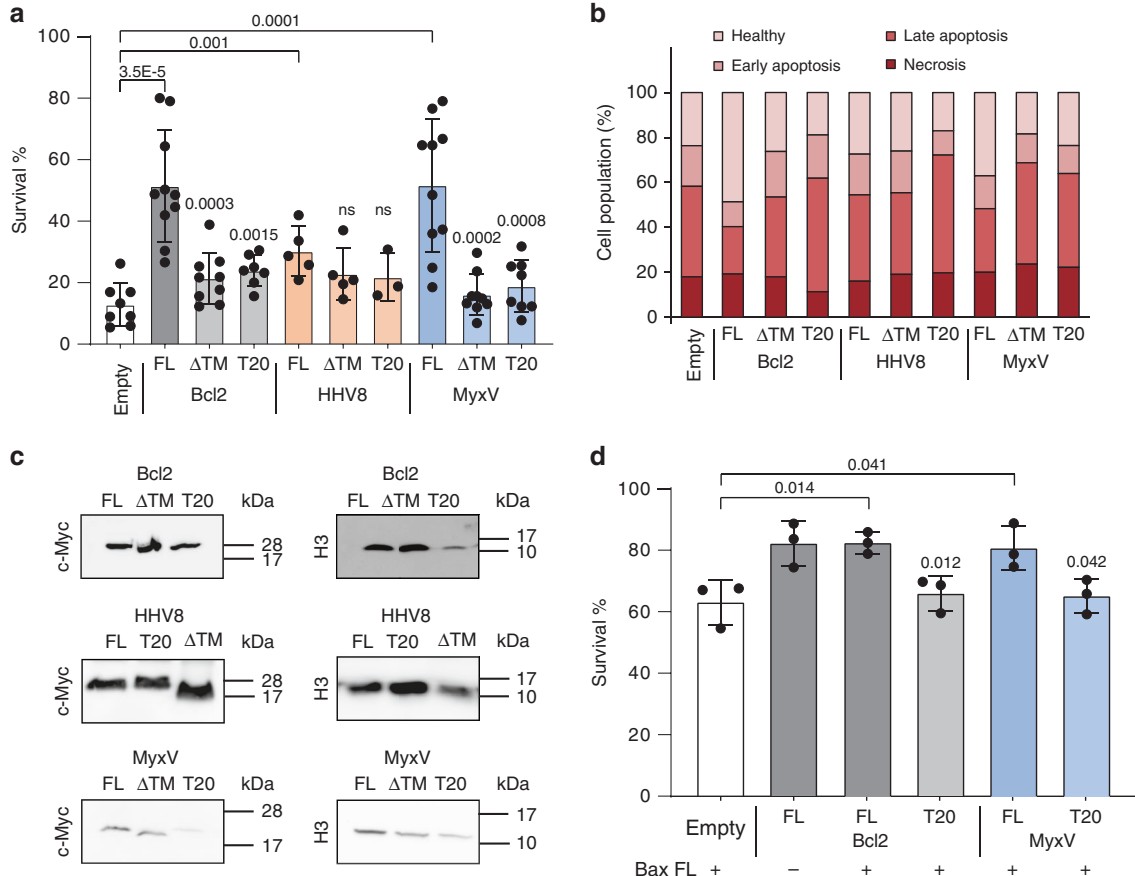

**Fig. 7 The role of vBcl2 TMD in apoptotic control. a** Role of vBcl2 TMD in control of doxorubicin-induced apoptosis. Cells were transfected with Bcl2, HHV8, and MyxV with (FL) or without the TMD (ΔTMD), or with the TMD substituted by the TMD of T20 (T20). Cells were then treated with doxorubicin and the percentage of surviving cells was calculated by Trypan blue staining. The survival percentage mean and standard deviation of at least three independent experiments are shown ($n \geq 3$). Solid dots represent the results of individual experiments. Transfection with an empty plasmid (Empty) was used as a negative control. Statistical differences are based on a two-tailed homoscedastic $t$-test ($p$-values are indicated above the corresponding bar, ns non-significant). **b** Alternatively, the percentage of cells in healthy, early apoptotic, late apoptotic, or necrotic states was measured by flow cytometry using propidium iodide staining and phosphatidyl serine labeling (FITC-Annexin V). The average percentage of cells in each of the aforementioned populations is shown. **c** Western blot analysis of protein levels. Histone 3 (H3) was used as a loading control ($n = 3$). **d** Role of MyxV TMD in Bax-induced apoptosis. Cells were transfected with Bcl2 and MyxV (FL) or with the TMD substituted by the TMD of T20 (T20) and co-transfected with (+) or without (−) Bax-FL as an apoptosis stimulus. Survival percentage mean and standard deviation of three independent experiments are shown ($n = 3$). Solid dots represent the results of individual experiments. Statistical differences are based on a two-tailed homoscedastic $t$-test ($p$-values are shown above the corresponding bars).

and host TMDs in the outer mitochondrial membrane are required for the control of apoptosis in a viral infection scenario.

## Discussion

To prevent the premature death of host cells, many viruses have acquired the ability to modulate apoptosis. As a result, the virus can replicate longer and circumvent activation of the immune system[18,50]. This infection-induced effect can be permanent, provoking the cell to persistently escape programmed cell death despite paracrine or autocrine stimulus[51]. Indeed, 8–16% of new cancer cases are attributable to carcinogenic infections[52,53]. Preventing acute infection episodes as well as the long-lasting effects associated with apoptosis evasion (including cancer) requires a detailed understanding of how viruses control programmed cell death. Viruses employ multiple strategies to block apoptosis. One of the most notable, at least in DNA viruses, is the expression of cellular pro-survival Bcl2 analogs[20]. Remarkably, many structural elements of cBcl2, including a Ct hydrophobic region, are preserved in their viral replicas. Here, we explored the role of the vBcl2 Ct domain in protein function. More specifically, we

focused on its function as a membrane anchor, its contribution to host–host and viral–host PPIs, and its importance in the control of cellular apoptosis.

In this report, we demonstrated that the cellular machinery recognizes Ct regions of the tested vBcl2s as truly being TMDs and therefore inserts them into the membrane despite a low hydrophobicity score for some of these regions (BoHV, MyxV). Furthermore, a positive (suggesting no insertion) $\Delta G_{\text{pred}}$ value for the insertion into lipid bilayers was predicted for VacV Ct region. Nonetheless, this predicted penalty did not preclude VacV Ct insertion. In fact, all the $\Delta G_{\text{exp}}$ values obtained in vitro were below −1 kcal/mol, suggesting a strong membrane insertion capacity.

Permeabilization of the MOM, considered in most cases the point of irreversible commitment to apoptosis, is controlled by interactions among pro- and anti-apoptotic Bcl2 members[54]. Most members of the Bcl2 family, including BH3-only proteins, bear a Ct TMD necessary for both their subcellular localization and apoptotic activity. Additionally, Bax and Bcl2 TMDs have previously been found to interact in biological membranes[17]. This

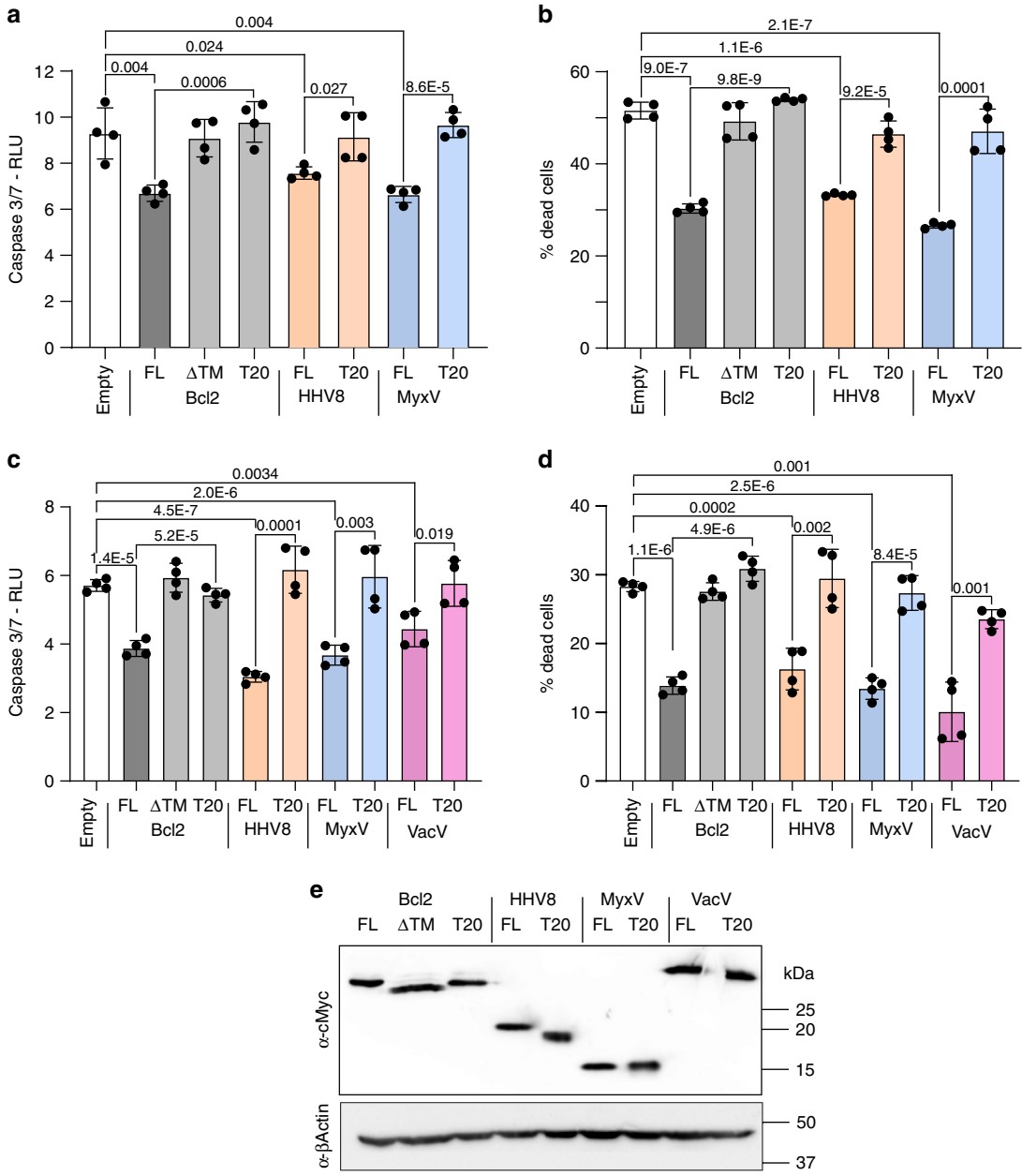

**Fig. 8 The role of viral TMDs on viral-induced apoptosis control. a**, **b** HeLa cells were co-infected with VacV-PKR and VT7LacOI (MOI 2.5 for each virus) and transfected with 50 ng of the plasmids Bcl2-FL, Bcl2-ΔTMD, Bcl2-T20, HHV8, HHV8-T20, MyxV, and MyxV-T20 or with an empty plasmid. IPTG was then added to induce protein expression. At 24 h.p.i, apoptosis was measured from duplicate samples using the Caspase-Glo 3/7 assay kit (**a**) or by measuring cell viability with Trypan blue staining (**b**). The mean and standard deviation of four technical replicates from two independent experiments are shown. Solid dots represent the results of individual replicates. Cells transfected with an empty plasmid were used as a control. Statistical differences are based on a two-tailed homoscedastic *t*-test, *p*-values are indicated above the corresponding bar. **c**, **d** HeLa cells were co-infected with VT7LacOI and MVA-C-ΔF1L deletion mutant (MOI 2.5 of each virus) and transfected with 50 ng of the plasmids Bcl2-FL, Bcl2-ΔTMD, Bcl2-T20, HHV8, HHV8-T20, MyxV, MyxV-T20, VacV, and VacV-T20 or with an empty plasmid. IPTG was then added to induce protein expression. At 24 h.p.i, apoptosis was measured from samples using the Caspase-Glo 3/7 assay kit (**c**) or by measuring cell viability by Trypan blue staining (**d**). The mean and standard deviation of four technical replicates from two independent experiments are shown. Cells transfected with an empty plasmid were used as control. Statistical differences are based on a two-tailed homoscedastic *t*-test, *p*-values are indicated above the corresponding bar. **e** Western blot analysis of Bcl2, HHV8, MyxV, and VacV with (FL) or without the TMD (ΔTMD) or with the TMD of T20. All viral proteins included a c-Myc tag. Beta-actin was used as a loading control (*n* = 3).

information, coupled with the newly discovered membrane-spanning capacity of the vBcl2 Ct hydrophobic region, led us to explore the homo-oligomerization potential of the newly identified vBcl2 TMDs. Our findings using two different complementation assays, i.e., BiFC and BlaTM, suggest that all vBcl2 TMDs can form intramembrane homo-oligomers. Sequence-

specific TMD oligomerization is influenced by lipid bilayer properties and thus its composition[55–57]. Our results indicate that the BiFC-observed homo-oligomers can form in vivo at the ER and mitochondrial membranes. It is particularly interesting to find these anti-apoptotic homo-oligomers at the mitochondrial membrane, the site of the pro-apoptotic action of Bax, Bak, and

other pro-apoptotic Bcl2 proteins[58]. These results not only indicate a similar distribution for vBcl2 and cBcl2 but also raise the possibility of a potential dynamic intracellular localization for vBcl2.

Despite some drawbacks, such as an elevated number of replicates needed or difficulty automating the process, we believe that the BiFC-based approach is particularly well suited for our study. The BiFC assay not only allowed us to conduct our experiments in eukaryotic cells but also replicated the native membrane topology of the vBcl2 proteins. Proteins with no signal peptide and a single TMD in their Ct end (known as tail-anchored proteins), such as the Bcl2 family, are a minority within the cellular membrane proteome (3–5%)[59]. By locating the TMD at the Ct end of our chimeras, we replicated its natural location/orientation and any potential for TMD–TMD interaction. Furthermore, the BiFC assay facilitated the study of hetero-oligomerizations and the identification of the subcellular compartment where the interactions occur.

Using BiFC, we also analyzed the potential TMD–TMD interactions between vBcl2 and cBcl2. Our results revealed a previously overlooked host–pathogen intramembrane interaction network. It should be noted that intramembrane oligomers can be fragile because of the small interaction surface between monomers[44,60–62] This history indicates that at least in some cases, the contact between the cellular and viral proteins occurs at the same time through soluble and membrane-spanning regions.

Of interest, all the viral TMDs we tested here, except that of BoHV, could interact with multiple cellular TMDs. The particularities of these connection circuits varied from virus to virus, potentially reflecting distinctive mechanisms of action. Indeed, a comprehensive PCA of the TMD–TMD interaction data revealed a similar interaction pattern for herpesviral HHV4 and HHV8 members on one hand, and poxviral VacV and OrfV members on the other. These similarities among closely related viruses might also suggest that the interaction network of each TMD is important for the survival of the virus so that it has been conserved throughout its evolution. Otherwise, the interaction pattern of each TMD would have been lost through the fast mutation rate and high evolutionary pressure scenario associated with the virus. Sequence similarity would have been expected to accompany similarities in the vBcl2 TMD interaction pattern. However, as the sequence alignment exposed, in the TMDs of the analyzed proteins, no sequence homology pattern could explain the observed results. This finding implies the presence of a structural pattern underlying the sequence that could govern the TMD–TMD interactions and that has been conserved throughout viral evolution. Of note, the similarities in the interaction network between closely related viruses also suggest that the observed interactions are not the result of random contacts among TMDs resulting from non-specific affinities or overcrowding.

Our assays allowed us to analyze each potential interaction independently. However, for each vBcl2 protein in vivo, all the TMD–TMD interactions with cBcl2s could occur at once. It remains to be confirmed whether there is a preferred (hierarchical) interaction that is responsible for the anti-apoptotic effect or there are several TMD–TMD interactions that occur side-by-side, all of them contributing to the cell death blockage. Currently, data exist to support either possibility. On the one hand, vBcl2s can interact with multiple cBcl2s. For example, as recently demonstrated, HHV4 drives chemoresistance and lymphomagenesis by inhibiting multiple cellular pro-apoptotic proteins[61]. On the other hand, a hierarchic interaction profile for the TMDs of cBcl2 has also been observed[63]. These two possibilities are not exclusive and most likely occur simultaneously in the cells.

According to our results, vBcl2 TMD are capable of forming homo- and hetero-oligomers. These two types of interactions might not be exclusive but collaborative/synergistic. The techniques we used for the study of hetero-oligomers (BiFC and BlaTM) do not impede the formation of homo-oligomers alongside them. In these assays, the formation of a heterodimer is required to obtain fluorescence (BiFC) or antibiotic resistance (BlaTM). Nonetheless, it is possible (even probable) that homo-oligomers are formed prior to the interaction between a viral and a cellular TMD. Indeed, VacV has been shown to adopt a dimeric fold through its soluble region which enables hetero-oligomeric binding to pro-apoptotic members of the Bcl2 family[64]. In this case, it is likely that the TMD first participates in the formation of the homodimer and then in the interaction with cBcl2.

Next, we focused on the TMD–TMD interaction between the strong partners Bcl2-HHV8 and Bax-MyxV. Bcl2 and HHV8 TMDs were detected in mitochondrial membranes where Bcl2 is thought to interact with pro-apoptotic members of the Bcl2 family to control apoptosis. Whether HHV8 requires its interaction with Bcl2 to exert its anti-apoptotic function remains to be seen. Our data also revealed that the presence of the soluble domains of Bcl2 did not affect TMD–TMD interaction between Bcl2 and HHV8 (Fig. 4a), reinforcing the idea that intramembrane interactions could occur side-by-side with contacts through soluble domains. With respect to the MyxV-Bax intramembrane interaction, we not only confirmed the interaction seen with the BiFC assay but also demonstrated that MyxV TMD can inhibit Bax TMD-induced apoptosis (Fig. 5). Furthermore, MyxV-FL but not MyxV-T20 could rescue Bax-FL-induced apoptosis. These results might imply that MyxV can exert its anti-apoptotic action through direct interactions with pro-apoptotic Bax (in agreement with the previous results[43]) and that its TMD is a key domain that plays an active role in the control of cell death. Nonetheless, other scenarios in which a third protein is involved cannot be ruled out. Of interest, the in silico analysis of both the putative HHV8-Bcl2 and the MyxV-Bax intramembrane dimers revealed ridge–groove arrangements between small and large residues in opposing TMDs. This type of interaction has been observed previously in other TMD dimers, particularly in the case of the GpA homodimer.

Finally, our results, using both drug and viral apoptotic stimuli, indicated that the TMD in the vBcl2s is necessary for the control of apoptosis. Furthermore, when HHV8, MyxV, or VacV lose their ability to interact with cellular proteins through the TMD, they become incapable of regulating apoptosis, as demonstrated when the TMD in the vBcl2 was deleted or substituted by the TMD of T20 (Figs. 7 and 8). These results open the way to the design of new antivirals that could interfere with these hosts–viral TMD–TMD interactions. Of interest, MyxV-FL behaved as a strong apoptosis inhibitor regardless of the stimuli. In contrast, HHV8-FL worked better as an apoptosis inhibitor when PKR (under a VacV promoter) was used as the stimulus. This divergence between MyxV and HHV8 suggests a difference in their mechanism of action, which could also be inferred by analyzing the interaction profile of their TMDs (Fig. 6 and Supplementary Fig. 10).

In conclusion, we have identified the Ct hydrophobic region of the vBcl2 as a true TMD that can interact with cBcl2 TMDs. We also have demonstrated that these intramembrane interactions are crucial for the viral control of cell fate. This work advances our understanding of how viruses control cellular apoptosis for their advantage and how apoptosis is regulated in the cell. Furthermore, the interactions described here expand knowledge about how viruses interact with their host.

## Methods

**Cell cultures, plasmids, and reagents**. Human embryonic kidney 293T cells (HEK 293T), human colorectal carcinoma-derived cells (HCT 116), and human epithelial cervical cancer cells (HeLa) were cultured in Dulbecco's modified Eagle's medium (DMEM) (Gibco) supplemented with 10% fetal bovine serum (FBS) (Gibco), and penicillin–streptomycin (P/S) (100 U/mL) (Gibco). All cells were grown at 37 °C, 5% $CO_2$.

The TMD sequences were synthesized by Invitrogen (GeneArt gene synthesis), PCR amplified, and subcloned into the appropriated vector either using a standard digestion-ligation protocol or using the InFusion cloning system following the manufacturer's protocol (Takara). Mutations into the TMD were introduced by site-directed mutagenesis using the Quick Change II kit following the manufacturer's instructions (Agilent Technologies). A full list of all the primers can be found in Supplementary Table 1. All DNA manipulations were confirmed by the sequencing of plasmid DNAs (Macrogen). Transfection of DNA into eukaryotic cells was performed in Opti-MEM reduced serum medium (Gibco) with Lipofectamine 2000 (Invitrogen) according to the manufacturer's specifications. The recommendations of the International Committee on Taxonomy of Viruses (http://www.ictvonline.org/index.asp) were used as guidelines for the viral nomenclature.

**In vitro transcription and translation**. The Lep-derived constructs were assayed using the TNT T7 Quick Coupled System (#L1170, Promega). Each reaction containing 1 μL of PCR product, 0.5 μL of EasyTag™ EXPRESS 35S Protein Labeling Mix (Perkin Elmer) (5.5 μCi), and 0.3 μL of microsomes (tRNA Probes) was incubated at 30 °C for 90 min. Samples were analyzed by SDS-PAGE. The bands were quantified using a Fuji FLA-3000 phosphoimager and the Image Reader 8.1 software. Free energy was calculated using: $\Delta G_{app} = -RT \ln K_{app}$, where $K_{app} = f2g/f1g$ being f1g and f2g the fraction of single glycosylated and double glycosylated protein, respectively. Endoglycosidase H treatment (Roche) was carried according to the specifications of the manufacturer.

**Bimolecular fluorescent complementation (BiFC) assay**. For the generation of BiFC chimeric plasmids including the Nt or Ct of the Venus Fluorescent Protein (VN, VC, respectively) Addgene #27097, #22011 (a gift from Chang-Deng H)[65] plasmids were modified to clone the cellular and viral Bcl2 TMDs at the Ct of the VFP. Chimeras (500 ng VN + 500 ng VC) were transfected into $2 \times 10^5$ HEK 293T cells together with a plasmid expressing *Renilla luciferase* under the CMV promoter (pRL-CMV) (50 ng) for signal normalization. Additionally, for the competition assay, 500 ng of a plasmid encoding Bcl2 full-length or Bcl2 lacking the TMD were transfected. Cells were incubated at 37 °C, 5% $CO_2$ for 48 h, PBS washed and collected for fluorescence and luciferase measurements (Victor X3 plate reader). For the Renilla luciferase readings, we used the Renilla Luciferase Glow Assay Kit (Pierce, Thermofisher) according to the manufacturer's protocol. In each experiment, the fluorescence/luminescence ratio obtained with the GpA homodimer was used as a 100% oligomerization value and the rest of the values adjusted accordingly. All experiments were done at least in triplicates.

**BlaTM assay**. Competent *E. coli* BL21-DE3 cells were co-transformed with N-BLa and C-BLa plasmids, version 1.1[39], containing a given TMD pair and grown overnight at 37 °C on LB-agar plates containing 34 μg/mL of chloramphenicol (Cm) and 35 μg/mL of kanamycin (Kan) for plasmid inheritance. After o/n incubation at 37 °C, colonies were either picked for immediate use or the plates were sealed with Parafilm (Pechiney Plastic Packaging) and stored at 4 °C for up to one week. Overnight cultures were conducted by inoculating 5 mL of LB-medium (Cm, Kan) with 10 colonies from one agar plate, followed by o/n incubation in an orbital incubator at 37 °C, 200 rpm. An expression culture was started with a 1:10 dilution of the overnight culture in 4 mL expression medium: LB-medium (Cm, Kan) containing 1.33 mM arabinose. After 4 h at 37 °C, the expression cultures were diluted to an $OD_{600} = 0.1$ in expression medium. To expose the bacteria to different ampicillin concentrations, an $LD_{50}$ culture was prepared by pipetting 100 μL of the diluted expression culture into each cavity of a 96-deep well plate (96 square well, 2 mL, VWR) containing 400 μL of expression media (final $OD_{600} = 0.02$). Freshly prepared ampicillin stock (100 mg/mL in ethanol) was added, resulting in ampicillin concentrations ranging from 0 to 350 μg/mL, depending on the affinity of the TMD under investigation. As a rule, the maximum ampicillin concentration to be used for a particular case should be about twice the mean $LD_{50}$. The plates were incubated in a moisturized container for 16 h at 37 °C and 250 rpm on a shaker (shaking amplitude 10 mm, KS 260 Basic, IKA) containing tips in every well to ensure a proper agitation. Cell density was measured via absorbance at 544 nm in a microplate reader (Victor X3, Perkin Elmer). To minimize clonal variation, at least two transformations were done and at least two separate $LD_{50}$ cultures were inoculated from each batch of transformed bacteria using ten colonies for each culture. Thus, at least 40 colonies entered each determination of $LD_{50}$. To measure and collect $LD_{50}$ values from the dose–response curves, we used Prism 8 from GraphPad.

To analyze the expression levels of the chimeras, competent *E. coli* BL21-DE3 cells were transformed with one N-BLa or C-BLa plasmid, version 1.1, containing a given TMD and grown overnight at 37 °C on LB-agar plates containing 34 μg/mL

of Cm (for N-BLa) or 35 μg/mL of Kan (for C-BLa) for plasmid inheritance. After o/n incubation at 37 °C, cultures were conducted by inoculating 5 mL of LB-medium (Cm or Kan) with 10 colonies from one agar plate, followed by o/n incubation in an orbital incubator at 37 °C, 200 rpm. An expression culture was started with a 1:10 dilution from the overnight culture in 4 mL expression medium: LB-medium (Cm or Kan) containing 1.33 mM arabinose. After 4 h at 37 °C, the expression cultures were diluted to an $OD_{600} = 0.1$ in 5 ml of expression medium (final volume) and grown o/n, 37 °C, 200xg. The morning after, 100 μL from each culture were transferred to a black 96-well plate to measure the fluorescence (Victor X3, Perkin Elmer).

**Cell-viability assays**. To measure doxorubicin-induced apoptosis $1.5 \times 10^6$ HeLa cells were plated in a 24 wells plate containing 0.5 ml of media in each well. After overnight incubation, each well was transfected in triplicates with 500 ng of DNA. After two days of expression, cells were treated with doxorubicin (stock 2 mM in DMSO) achieving a final concentration of 15 μM. Approximately, 16 h post-treatment cells (including those in the supernatant) were collected and their viability was measured using Trypan blue and using an automated cell counter (Invitrogen, Countess™ II). At least, 2 measurements per well were done. Additionally, the viability and apoptotic state of the cells was measured using the BD Annexin V: FITC Apoptosis Detection Kit II according to the manufacturer specifications and a BD LSR-Fortessa (Beckton Dickinson). All flow cytometry countings were done at the tissue culture and flow cytometry core facilities of the SCSIE (University of Valencia) following the recommendations of the BD Annexin V: FITC Apoptosis Detection Kit II. To measure Bax TMD-induced apoptosis HCT 116 cells were transfected with 250 ng of the plasmid encoding VN-Bax and 250 ng of either VC-H2, VC-Bcl2 or VC-MyxV encoding plasmids or mock-transfected. After 48 h, cells were trypsinized and viability was measured twice per well using Trypan Blue and automated Countess™ II cell counter (Invitrogen). The percentage of survival was calculated using the mock-transfected cells as the reference value.

To measure Bax-induced apoptosis, $3 \times 10^6$ HeLa cells were plated in a 24-well plate (0.5 ml of media in each well). After overnight incubation, cells were transfected (in triplicate) with 500 ng of either MyxV-FL, M11L-T20, Bcl2-FL, Bcl2-T20, or empty pCAGGS (negative control). Additionally, cells were co-transfected with 400 ng of Bax DNA. For the positive control, 500 ng of Bcl2-FL and 400 ng of empty pCAGGS were transfected. After 24 hrs of expression, cells were collected and their viability was measured using Trypan blue and an automated cell counter (Invitrogen, Countess™ II). At least 2 measurements were done per well.

**Bio-informatic resources**. Prediction of the TMD on full-length sequences was done either with the ΔG prediction server v1.0[31,32] or the TMHMM server v2.0[29,30] using default parameters. Principal Component Analysis (PCA) was done with the Gene Expression Similarity Investigation Suite (Genesis)[66] (http://genome.tugraz.at) using the default parameters. For the multiple sequence alignment, *Clustal Omega* was launched from the EMBL-EBI site[67]. Contact surface predictions for TM dimers were done with PREDDIMER[67] (http://model.nmr.ru/preddimer/). From all the options provided by the algorithm, only the lowest energy option was considered.

**Confocal microscopy**. Confocal micrographs were done at the Microscopy Core Facility of the SCSIE (University of Valencia) using an Olympus FV1000 confocal microscope with a ×60 oil lens. ER (Sec61β), plasma membrane (the first twenty residues of neuromodulin) and, mitochondria (Tomm20) mCherry fluorescent-labeled markers were obtained from Addgene plasmid repository #49155[68], #55779[69] and #55146 (a gift from Michael Davidson, Institute of Molecular Biophysics and Center for Materials Research and Technology, The Florida State University) respectively. Cells ($5 \times 10^3$ cells/well) were seeded on 10 mm coverslides treated with poly-Lys and placed in 24-well plates. The next day, cells were transfected with the appropriate plasmids. After 24 h, the cells were fixed (4% paraformaldehyde) and DAPI stained before image capture. A 1:1000 dilution in TBS 0.005% Tween Rabbit anti c-Myc (Sigma PLA0001) antibody followed by an anti-Rabbit Alexa 488 conjugated (Life Technologies A21206) (1:1000) was used to label viral proteins. Pictures were taken in an Olympus FV1000 confocal microscope. Laser intensity was individually adjusted in all samples. Pictures were not used for quantification.

**Viruses and infections**. The VacV recombinants used in this study have been previously described: VacV-PKR (previously named as VV-p68)[48], which expresses the dsRNA-dependent protein kinase upon induction of the isopropyl β-D-thio-galactosidase (IPTG)-dependent inducible promoter, VT7LacOI that expresses the T7 RNA polymerase in an IPTG-dependent manner, the parental MVA-C and the deletion mutant MVA-C-ΔF1L lacking the viral F1L gene[49]. To express viral and cBcl2 proteins alongside viral infection, the selected DNA sequences were cloned into pVOTE.2[70] plasmid under the control of the IPTG-dependent inducible T7 promoter. Infections were performed on pre-confluent (<75%) cell monolayers. Cultures were co-infected with VT7LacOI and either VV-PKR, MVA-C, or MVA-C-ΔFIL diluted in DMEM to a multiplicity of infection (MOI) of 2.5 PFU/cell of each virus. After the one-hour adsorption cells were transfected with 50 ng of the

indicated plasmid. Transfections of infected cells were performed with Lipofecta-mine 2000 reagent (Invitrogen), following the protocol recommended by the supplier. Infected cells were incubated in the presence of IPTG (5 mM) at 37 °C until 24 h.p.i.

**Caspase 3/7 activity assay**. Quantification of caspase activity was carried out by using the Caspase-Glo 3/7 assay kit (Promega) following the protocol recommended by the supplier. HeLa cell monolayers grown in 24-well plates were infected and transfected, in duplicate, under the conditions indicated above. At 24 h.p.i. cells were harvested in medium and kept frozen until their analysis. 25 µL of the cell lysates under study was added to 25 µL of Caspase-Glo 3/7 reagent in a 96-well plate. Plates were gently shaken and then incubated in the dark at room temperature for 60 min before recording the luciferase activity by using an Appliskan luminometer (ThermoScientific).

**Western blot analysis**. Cell monolayers were lysed in Laemmli´s sample buffer (62.5 mM Tris-HCl [pH 6.8], 2% sodium dodecyl sulfate [SDS], 0.01% bromo-phenol blue, 10% glycerol and 5% β-mercaptoethanol). Protein samples were subjected to 12% SDS-polyacrylamide gel electrophoresis (PAGE) and transferred to nitrocellulose membranes (BioRad). Membranes were blocked for 30 min at room temperature in Tris-buffered saline supplemented with 0.05% Tween 20 (TBS-T) containing 5% non-fat dry milk, and later incubated with primary anti-bodies diluted in the same buffer at 4 °C overnight. Antibodies used in this study were β-actin (Santa Cruz Biotechnology SC-47778), c-Myc (Sigma PLA0001 or Roche 11667149001), Histone 3 (Sigma H0164), and Flag (Sigma B3111). Then, membranes were washed with TBS-T and incubated with goat anti-mouse IgG-peroxidase conjugate (Sigma DC02L) for 1 h at room temperature and washed again. All antibodies were used at a 1:10,000 dilution in TBS-T with 5% non-fat dry milk. Detection of immunoreactive proteins was carried out using the enhanced chemiluminescence (ECL) reaction (SuperSignal ThermoScientific) and detected by the ChemiDoc Touch Imaging System (BioRad).

**Reporting summary**. Further information on research design is available in the Nature Research Reporting Summary linked to this article.

## Data availability
The data that support the findings of this study are available from the corresponding author upon reasonable request. Source data are provided with this paper.

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

## Acknowledgements

We thank the Generalitat Valenciana (GV/2016/139 Grupos emergentes program and PROMETEO/2019/065 Prometeo Program) and the Spanish Ministry of Economy and Competitiveness (MINECO) (Grant Nos. BFU2016-79487 and AGL2017-87464-C2-1-P). G.-D. and E.D.-B. are the recipients of a predoctoral grant from the Spanish Ministry of Science, Innovation and, Universities (MICINN) (FPU18/05771 and FPU18/01873 respectively). B.G. is the recipient of a predoctoral grant from the University of Valencia (Atracció de Talent Program). We thank B. Perdiguero and M. Esteban for the MVA-C-ΔF1L virus.

## Author contributions

L.M.-G., I.M., and D.R. designed the experiments. M.J.G.-M., G.D., B.G., E.D.-B., and L.M.-G. performed the experiments. All authors contributed to the analysis of the data. L.M.-G. wrote and edited the manuscript.

## Competing interests

The authors declare no competing interests.
