## [Peer Review File · Nature Communications]

REVIEWER COMMENTS

Reviewer #1 (Remarks to the Author):

This is a nice biochemical study where the authors predict the presence of virus Bcl2 TM analogs among viral encoded proteins in pox and herpesviruses. Expression of these proteins results in membrane localization, interactions with Bcl2 and inhibition of apoptosis during viral infection. While the results are clear and rigorous, they are based only in protein overexpression, and these proteins are not studied for function in the context of its own viral infection. The authors need to show phenotypes of virus mutants in such proteins in order to increase the significance of their studies and to validate their function in the context of viral infection

In addition, the authors mention that to their knowledge this is the first report of viral-host intramembrane protein interactions. This is not the case. As an example, see this review doi: 10.1146/annurev-micro-091313-103727

Reviewer #2 (Remarks to the Author):

The manuscript is well written, makes a compelling case for interactions between transmembrane domains (TMDs), and the conclusions are largely supported by the data. However, there are several issues that need to be addressed:

1. Why are there significant differences in the readout when using either the N or the C-terminal fragment in the VFP assay (see fig 3).
2. Topologically, Vac F1L was shown to be a domain swapped dimer (see e.g. Campbell et al JVI 2014), however in the TMD assays it is used effectively as a monomer. How would dimeric vBcl-2 topology impact on homo and heterooligomerization of the TMD, and the proposed mechanism? This should be discussed, particularly in light of the statement on page 17 line 24: Our data also revealed that the presence of the soluble domains of Bcl2 did not affect TMD–TMD interaction between Bcl2 and HHV8 (Fig. 4A), reinforcing the idea that intra-membrane interactions could occur side-by-side with contacts through soluble domains.
3. Page 15 line 9 - Interactions between cBcl2 proteins require the membrane. This is not correct as stated and overstates the importance of the membrane. BH3 only proteins can interact with cellular (and indeed viral) Bcl-2 proteins in the absence of a membrane, as shown by numerous studies using recombinant anti-apoptotic Bcl-2 proteins with either full length BH-3 only proteins or peptides encoding the BH3 motif as measured by SPR, ITC, fluorescence polarization or many other methods. Whilst the membrane is undoubtedly important, it is not in the absolute term as is stated here.
4. Page 18 line 3. The authors state that “In respect to the MyxV-Bax intramembrane interaction, we not only confirmed the interaction seen with the BiFC assay but also demonstrated that MyxV TMD can inhibit Bax TMD-induced apoptosis (Fig. 5). This result might imply that MyxV can exert its anti-apoptotic action through direct interactions with pro-apoptotic Bax, in agreement with previous results”. It has previously been shown that M11L confers protection against Bax mediated apoptosis, and that a mutation in the conserved M11L binding groove that reduces Bax binding leads to a loss of protection against Bax mediated apoptosis (Kvansakul et al 2007, Mol Cell). These data suggested

at the time that the interaction is driven by an in-groove dependent interaction. Have the authors considered using vBcl-2 mutants bearing in groove mutations in their assays to attempt to delineate the influence of in-groove interactions compared to their observed TMD dependent effect? Such experiments would be critical to evaluate the role of the TMD interactions and the hierarchy of drivers for apoptosis controlling PPIs.

Minor issues:

Figure legend:

Supp Fig 8 – that would be dimer, not dimmer

Supp Fig 3 – chimeras instead of quimeras (twice)

Supp Fig 6 – which statistical test was used?

Fig 3, Supp Fig 9, supp Fig 11, : How was the significance of the differences evaluated to be statistically relevant?

Fig 4: Either Bcl2 or BCL2 not BCL2. Also putative dimer between Bcl2 TMD and HHV8 TMD

Fig 5: Model of a putative dimer between MyxV TMD and Bax TMD

Fig. 6 : The colored areas highlight

Reviewer #3 (Remarks to the Author):

vBcl-2s are important regulators of apoptosis during infection with large DNA viruses. The mitochondrial localization is being appreciated as relevant to the regulation of apoptosis. This study provides evidence that the C-terminal membrane anchors of vBcl-2 proteins form homodimers as well as heterodimers with cBcl-2s, and that this dimerization is relevant to the inhibition of apoptosis. Although the localisation to mitochondria is now appreciated, this issue of dimer-formation has been hardly explored at all. I feel that this is an important issue, and the authors use a number of sophisticated assays in independent approaches to address the question of the role of the TMDs of vBcl-2s. The data are mostly convincing and well presented. The following are issues that in my view need to be addressed however:

1. The membrane insertion experiment shown in Fig. 1 should be better controlled. Although it seems plausible it is not actually shown that the three bands represent the various glycosylation states. A control with a sequence of similar length that is not predicted to insert should be shown.

2. Although it has been shown before that the Bax-TMD can induce apoptosis, this is very likely not the way Bax kills normally. During apoptosis, Bax-insertion into the outer mitochondrial membrane follows a much more complicated process. Because this study is also about the physiological activity

I think it should also be tested, whatever the result will be, whether for instance MyxV (Fig. 5C) can also inhibit apoptosis induced by full-length Bax.

3. The statistics used are not always clear. It has in each figure to be stated how often an experiment was done, and whether for instance triplicates are triplicates within one experiment or are meant to indicate independent repeats. The level of significance and the clearer description of the statistical test (which form of the t-test for instance?) have to be given, e.g. in Fig. 2B.

4. The glycine mutation (Fig. 4) may alter membrane insertion and protein stability. This has to be controlled. It would also be good to have a functional test of this mutant, for instance in an experiment like Fig. 8.

5. The experiment shown in Fig. 4a has to be better described. I can nowhere find information which cells were used or how much, or how, Bcl-2 was added, or whether the expression levels of the Bcl-2-variants were tested.

6. In Fig. 8 it would be good to have some more objective measure on the amount of apoptosis induced, rather than the relative amounts of caspase-substrate cleavage. Please provide for instance number of dead cells in infected vs. uninfected cells.

7. I am not saying this is essential but the virus-induced apoptosis (Fig. 8) would be more convincing if a virus without its vBcl-2 were used. The VacV used would still express F1, which however appears to be overwhelmed by pro-apoptotic factors. This is somewhat different from the normal situation where virus-expressed vBcl-2 actually inhibits apoptosis.

Minor

Please put in the gates used to determine the sizes of the various populations in the FACS blots in Fig. 7.

Expression of the Bla-chimaeras is measured by a fluorescence assay. What was actually measured (Fig. S4)?

I am not sure that the point should be made of a 'dynamic intracellular localization' of vBcl-2s (p. 8). For apoptosis inhibition they almost certainly have to be in the mitochondrial membrane, and there may well be factors other than the TMD to determine localization of the full-length protein. Please state what the cellular markers to look for subcellular localization were (rather than just giving AddGene numbers).

Suppl. Figure 8: b is MyxV, not also HHV8 as said in the legend.

What does a negative intramembrane interaction (Fig. S11) mean?

Reviewer #1 (Remarks to the Author):

This is a nice biochemical study where the authors predict the presence of virus Bcl2 TM analogs among viral encoded proteins in pox and herpesviruses. Expression of these proteins results in membrane localization, interactions with Bcl2 and inhibition of apoptosis during viral infection. While the results are clear and rigorous, they are based only in protein overexpression, and these proteins are not studied for function in the context of its own viral infection.

We thank the reviewer for their complements, helpful feedback and suggestions to expand upon our findings. Responses are provided below each individual comment. Changes in the manuscript have been highlighted in **red**. Please note that new supplementary figures have been added and figure numbers have changed accordingly.

1. The authors need to show phenotypes of virus mutants in such proteins in order to increase the significance of their studies and to validate their function in the context of viral infection.

As suggested by the reviewer, to further explore the role of vBcl2 TMD interactions in the control of apoptosis, we added a Modified Vaccinia Ankara virus mutant lacking F1L protein (MVA- Δ F1L) to our study. F1L has been proposed as the only orthopoxvirus Bcl2 homolog that directly inhibits the Bak/Bax checkpoint^{1,2}. Thus, MVA- Δ F1L provides a viral background in which no vBcl2 is expressed. While replication of the MVA- Δ F1L is similar to its parental virus, the deletion of F1L enhances infection-induced apoptosis, removing the need to express PKR as an apoptotic stimulus. Additionally, MVA- Δ F1L affords us the possibility of studying the role of F1L TMD interactions.

The results obtained with MVA- Δ F1L have been added to figure 8 and to the *Results* section (page 15, line 24). These results reveal the importance of the vBcl2 TMD interactions to be greater than previously observed.

2. In addition, the authors mention that to their knowledge this is the first report of viral-host intramembrane protein interactions. This is not the case. As an example, see this review doi: 10.1146/annurev-micro-091313-103727

We apologize for the oversight and thank the reviewer for pointing out this article. We have consequently removed the above mentioned statement from the manuscript.

Reviewer #2 (Remarks to the Author):

The manuscript is well written, makes a compelling case for interactions between transmembrane domains (TMDs), and the conclusions are largely supported by the data. However, there are several issues that need to be addressed:

We are honored that the reviewer found our study to be well written and compelling. We thank the reviewer for their constructive comments on the manuscript and for their suggestions to expand upon our findings. Individual responses are provided below each comment. Changes in the manuscript have been highlighted in **red**. Please note that new supplementary figures have been added and figure numbers have changed accordingly.

1. Why are there significant differences in the readout when using either the N or the C-terminal fragment in the VFP assay (see fig 3).

The BlaTM approach was used to show that the reporter signal resulting from a TMD-TMD interaction depends not only on the sequences of the interacting TMDs, and thus their inherent affinity, but also on the orientation of the interacting surfaces of the TMDs in relation to their accompanying signaling domains³⁻⁵. Despite the presence of a linker in the BlaTM chimera³.

We believe that a similar phenomenon can occur when using the BiFC assay. To demonstrate our hypothesis, we chose the interaction between Bcl2 and BoHV TMDs, which was the first of the interactions where significant differences were found between the two possible BiFC combinations. While VN-BoHV/VC-Bcl2 came out in our screening as a TMD-TMD interaction, VN-Bcl2/VC-BoHV did not. However, insertion of one, two, or three residues at the Nt end of BoHV TMD significantly increased the observed RFUs above their controls (Supplemental Fig. 13) (page 12 line 8). Although this result confirms our hypothesis, due to the large number of interactions studied and despite a large number of replicas, we cannot rule out the presence of some false positives or false negative results.

2. Topologically, Vac F1L was shown to be a domain swapped dimer (see e.g. Campbell et al JVI 2014), however in the TMD assays it is used effectively as a monomer. How would dimeric vBcl-2 topology impact on homo and heterooligomerization of the TMD, and the proposed mechanism? This should be discussed, particularly in light of the statement on page 17 line 24: Our data also revealed that the presence of the soluble domains of Bcl2 did not affect TMD-TMD interaction

between Bcl2 and HHV8 (Fig. 4A), reinforcing the idea that intra-membrane interactions could occur side-by-side with contacts through soluble domains.

This is a very interesting concept and we are glad that the reviewer brought it up. The assays used in our work to study TMD-TMD hetero-oligomers interactions do not impede the formation of homo-oligomers. In these assays, the formation of a hetero-dimer is required to obtain fluorescence (BiFC) or antibiotic resistance (BlaTM). Nonetheless, it is possible (and probable) that TMD homo-oligomers are formed alongside hetero-oligomers.

Our assays cannot identify the functional unit for TMD-TMD interactions (monomers, dimers...), and, as the reviewer suggests, it could be a dimer. In the case of VacV F1L, formation of a TMD homo-dimer could be necessary for the intramembrane interaction with Bcl2 or Bax TMD. We acknowledge the importance of this concept and accordingly have devoted a new paragraph in the *Discussion* section to it (page 20 line 4).

3. Page 15 line 9 - Interactions between cBcl2 proteins require the membrane. This is not correct as stated and overstates the importance of the membrane. BH3 only proteins can interact with cellular (and indeed viral) Bcl-2 proteins in the absence of a membrane, as shown by numerous studies using recombinant anti-apoptotic Bcl-2 proteins with either full length BH-3 only proteins or peptides encoding the BH3 motif as measured by SPR, ITC, fluorescence polarization or many other methods. Whilst the membrane is undoubtedly important, it is not in the absolute term as is stated here.

With this statement, we had intended to indicate the importance in apoptosis of the biological membrane and the interactions among members of the Bcl2 family that occur in the lipid bilayer. We agree with the reviewer that it was an oversimplification and have modified the text accordingly (page 17, line 12).

4. Page 18 line 3. The authors state that “In respect to the MyxV-Bax intramembrane interaction, we not only confirmed the interaction seen with the BiFC assay but also demonstrated that MyxV TMD can inhibit Bax TMD-induced apoptosis (Fig. 5). This result might imply that MyxV can exert its anti-apoptotic action through direct interactions with pro-apoptotic Bax, in agreement with previous results”. It has previously been shown that M11L confers protection against Bax mediated apoptosis, and that a mutation in the conserved M11L binding groove that reduces Bax binding leads to a loss of protection against Bax mediated apoptosis (Kvansakul et al 2007, Mol Cell).

These data suggested at the time that the interaction is driven by an in-groove dependent interaction. Have the authors considered using vBcl-2 mutants bearing in groove mutations in their assays to attempt to delineate the influence of in-groove interactions compared to their observed TMD dependent effect? Such experiments would be critical to evaluate the role of the TMD interactions and the hierarchy of drivers for apoptosis controlling PPIs.

As suggested by the reviewer, to explore the influence of MyxV in-groove mutations we first explored the anti-apoptotic potential of MyxV using full-length Bax as a stimulus. In our assay, both Bcl2 and MyxV FL proteins could rescue Bax induced apoptosis. However, the substitution of MyxV or Bcl2 TMD by the Ct membrane anchoring segment of T20 eliminated their anti-apoptotic properties (Figure 7). The *Results* and *Discussion* sections have been modified accordingly (page 14, line 16, and page 21, line 2).

Next, we compared the role of MyxV vBcl2 in-groove and TMD interactions on the protein anti-apoptotic functions using our newly developed assay. Our results indicated that, as suggested previously, in-groove mutations impair MyxV protection against Bax mediated apoptosis. However, the substitution of the TMD in the viral protein by T20 TMD has a stronger effect on the apoptotic control. These results have been included in Supplementary Fig. 16 and in the *Results* section (page 14, line 23).

As suggested by the molecular models we completed our study of the interaction between MyxV and Bax TMD by analyzing the role of the small residues within them, and their implication on a ridge-groove arrangement between both helices (Figure 5e) (page 11, line 5).

Minor issues:

Figure legend:

We have revised all figure legends. We hope that they are now clearer and more informative.

Supp Fig 8 that would be dimer, not dimmer.

The misspelling in S9 (previously S8) has been corrected.

Supp Fig 3 chimeras instead of quimeras (twice).

The misspelling in S4 (previously S3) has been corrected.

Supp Fig 6 which statistical test was used?

We used a two-tailed homoscedastic t-test to compare the oligomerization derived-fluorescence of any given interaction and its negative controls. This information has been added to the figure legend (S7, previously S6) (page 40, line 4).

Fig 3, Supp Fig 9, supp Fig 11: How was the significance of the differences evaluated to be statistically relevant?

We have revised the figure legends (pages 43 and 44) to clarify the methodology. In the experiments shown in Fig. 3, Supplementary Fig. 10 (previously S9), and Supplementary Fig. 12 (previously S11), we considered an interaction when its associated fluorescence values were significantly higher (two-tailed homoscedastic t-tests, $p\text{-value} < 0.05$) than those of the negative controls, X/T20 and T20/Y for the X/Y combination. That is, two t-tests were performed X/Y vs. X/T20 and X/Y vs. T20/Y, and for an interaction to be considered both p-values must have been below 0.05. The values in Supplementary Fig. 10 show the highest of these two p-values.

Fig 4: Either Bcl2 or BCL2 not BCl2. Also putative dimer between Bcl2 TMD and HHV8 TMD
The errors have been corrected.

Fig 5: Model of a putative dimer between MyxV TMD and Bax TMD
The error has been corrected.

Fig. 6 : The colored areas highlight
The error has been corrected.

Reviewer #3 (Remarks to the Author):

vBcl-2s are important regulators of apoptosis during infection with large DNA viruses. The mitochondrial localization is being appreciated as relevant to the regulation of apoptosis. This study provides evidence that the C-terminal membrane anchors of vBcl-2 proteins form homodimers as well as heterodimers with cBcl-2s, and that this dimerization is relevant to the inhibition of apoptosis. Although the localisation to mitochondria is now appreciated, this issue of dimer-formation has been hardly explored at all. I feel that this is an important issue, and the authors use a number of sophisticated assays in independent approaches to address the question of the role of the TMDs of vBcl-2s. The data are mostly convincing and well presented.

We thank the reviewer for their complements on our study. We are grateful for the constructive comments and suggestions to expand upon our findings. Responses are provided below each individual comment. Changes in the manuscript have been highlighted in **red**. Please note that new supplementary figures have been added and figure numbers have changed accordingly.

The following are issues that in my view need to be addressed however:

1. The membrane insertion experiment shown in Fig. 1 should be better controlled. Although it seems plausible it is not actually shown that the three bands represent the various glycosylation states. A control with a sequence of similar length that is not predicted to insert should be shown.

As suggested by the reviewer, to provide a better control for membrane insertion of vBcl2 TMDs, we have added a non-insertion/mono-glycosylated control with a sequence of similar length to the domains under investigation. For this purpose, we chose the Ct region of the BH3-only protein Noxa. Previous results have shown that this region does not insert into biological membranes ⁶.

Additionally, to ensure that the observed increase in molecular weight was due to glycosylation of the G1 and/or G2 acceptor sites, samples were translated in the presence or absence of Endoglycosydase H, a glycan-removing enzyme. These changes have been added to Fig. 1. Previous results including the Δ H2 control are now shown as well in Supplementary Fig. 3. Corresponding modifications to the *Results* section can be found on page 5, line 19. Please note that the experimental Δ Gs shown in Fig. 1 have been changed due to the increased number of replicates.

2. Although it has been shown before that the Bax-TMD can induce apoptosis, this is very likely not the way Bax kills normally. During apoptosis, Bax-insertion into the outer mitochondrial membrane follows a much more complicated process. Because this study is also about the physiological activity I think it should also be tested, whatever the result will be, whether for instance MyxV (Fig. 5C) can also inhibit apoptosis induced by full-length Bax.

As suggested by the reviewer, we explored the anti-apoptotic potential of MyxV using full-length Bax as a stimulus. In our assay, both Bcl2 and MyxV FL proteins could rescue Bax induced apoptosis. However, the substitution of the TMD in either Bcl2 or MyxV by the T20 Ct membrane anchoring segment eliminated their anti-apoptotic properties (Fig. 7). We have added the results of the new experiment to the *Results* and *Discussion* sections (page 14, line 16 and page 21, line 2).

Additionally, to gain further insight on the MyxV-Bax TMD-TMD viral-host interaction, we analyzed the role of small residues on the ridge–groove arrangement between both helices. The results of these experiments has been added to Fig. 5 and to the *Results* section (page 11, line 5).

3. The statistics used are not always clear. It has in each figure to be stated how often an experiment was done, and whether for instance triplicates are triplicates within one experiment or are meant to indicate independent repeats. The level of significance and the clearer description of the statistical test (which form of the t-test for instance?) have to be given, e.g. in Fig. 2B.

We apologize for the missing information and have amended the text to clarify the statistics. The type of statistical analysis used and the number of experimental and technical replicates has now been added, and we have carefully revised all figure legends.

4. The glycin mutation (Fig. 4) may alter membrane insertion and protein stability. This has to be controlled. It would also be good to have a functional test of this mutant, for instance in an experiment like Fig. 8.

As suggested by the reviewer, we have analyzed the membrane insertion of HHV8 G158L and Bcl2 G227L. For this purpose, we used the Lep assay described in Fig. 1. Glycine mutations did not alter membrane insertion or protein stability (Supplementary Fig. 3) (page 10, line 3). Additionally, we tested the membrane insertion of any other hydrophobic region with amino acid substitutions (Supplementary Fig. 3) (page 11, line 10).

Next, we tested the ability of HHV8 G158L to control apoptosis after treatment with doxorubicin (see figure below). Our results showed no differences between HHV8 wt and HHV8 G158L in their ability to control apoptosis. Protein levels were analyzed by Western Blot. This result might indicate that HHV8 does not require an specific TMD-TMD interaction with Bcl2 to control apoptosis. On the other hand, although binding of HHV8 G158L TMD to Blc2 TMD is reduced, it is not completely eliminated and the residual interaction might be enough to prevent apoptosis (Fig. 4e). Please also note that HHV8 did not show a strong apoptotic inhibition when doxorubicin was used as a stimulus. Therefore, it is possible that our assay is not sensitive enough to detect small differences between HHV8 FL and HHV8 G158L variant.

This is an interesting, although preliminary result. As such, we have decided to not include it in the manuscript.

The role of HHV8 G158L in apoptotic control. **a**, Cells were transfected with HHV8 full-length wt (FL) with a substitution of Gly 158 by Leu (G158L), or with the TMD substituted by the TMD of Tomm20 (T20). Cells were then treated with doxorubicin and the percentage of surviving cells calculated by Trypan blue staining. Survival percentage mean and standard deviation of six technical replicates from two independent experiments are shown. **b**, Western blot analysis of protein levels. Histone 3 (H3) was used as a loading control.

5. The experiment shown in Fig. 4a has to be better described. I can nowhere find information which cells were used or how much, or how, Bcl-2 was added, or whether the expression levels of the Bcl-2-variants were tested.

As indicated by the reviewer, we have revised the figure legend (page 41) and the *Methods* section (page 23) for clarity.

6. In Fig. 8 it would be good to have some more objective measure on the amount of apoptosis induced, rather than the relative amounts of caspase-substrate cleavage. Please provide for instance number of dead cells in infected vs. uninfected cells.

As suggested by the reviewer, in order to have a more objective measure of the amount of apoptosis, we have repeated the experiment and measured the cell death rate with Trypan blue staining. The new results are now included alongside the caspase 3/7 measurements (Figs 8a and 8c). We have decided to show the percentage of dead cells in order to facilitate comparison with caspase 3/7 levels.

We have also included a dual measure of apoptosis (caspase 3/7 and cell death rate by Trypan blue staining) in the infections with MVA- Δ F1L (see comment 7).

7. I am not saying this is essential but the virus-induced apoptosis (Fig. 8) would be more convincing if a virus without its vBcl-2 were used. The VacV used would still express F1, which however appears to be overwhelmed by pro-apoptotic factors. This is somewhat different from the normal situation where virus-expressed vBcl-2 actually inhibits apoptosis.

We agree with the reviewer's assessment. As such, we have added infections with MVA- Δ F1L to our study. The results can be found in Fig. 8 and in the *Results* section (page 15 line 24). According to the results in the new experimental setting, we believe that the relevance of the vBcl2 TMD interactions is greater than previously observed.

Minor

Please put in the gates used to determine the sizes of the various populations in the FACS blots in Fig. 7.

We have added the gates to the FACS blots. Please note that the data has been moved to Supplementary Fig. 14.

Expression of the Bla-chimaeras is measured by a fluorescence assay. What was actually measured (Fig. S4)?

The BlaTM assay was designed to identify TMD oligomerization through the reconstitution of the β -lactamase and it allows for easy monitoring of protein levels thanks to the GFP at the Ct end of the chimeras (Supplementary Fig. 5a). In Supplementary Fig. 5b (previously S4) we measured GFP fluorescence levels to ensure similar expression levels of the chimeras. A detailed explanation of the protocol can be found in the *Methods* section (page 25, line 5). We have modified the figure legend to clarify the results presented (page 39).

I am not sure that the point should be made of a 'dynamic intracellular localization' of vBcl-2s (p. 8). For apoptosis inhibition they almost certainly have to be in the mitochondrial membrane, and there may well be factors other than the TMD to determine localization of the full-length protein.

Bcl2 proteins control the mitochondrial membrane permeabilization and apoptosis by directly localizing to this organelle. However, current knowledge related to their subcellular distribution indicates that they are also found in other intracellular compartments such as the ER, the Golgi apparatus, the nucleus, and the peroxisomes where they not only remotely regulate mitochondrial membrane permeabilization but also participate in major cellular processes including calcium homeostasis, cell cycle control, and cell migration. Furthermore, it has been shown that the

distribution of Bcl2 proteins inside the cell is a dynamic process profoundly affected by changes in the cellular microenvironment.

We agree with the reviewer that factors other than the TMD determine the localization of the full-length protein. Nonetheless, we believe that the observed subcellular distribution of the vBcl2-derived chimeras suggest a multi-organelle distribution and, potentially, a dynamic localization. To clarify our thoughts on this matter, we have modified our statement in the *Results* section (page 8, line 5) and expanded our thoughts on the dynamic behavior of vBcl2s, and moved these remarks to the *Discussion* (page 18, line 1).

Please state what the cellular markers to look for subcellular localization were (rather than just giving AddGene numbers).

The nature of the subcellular markers has been added to the *Methods* section (page 27, line 7). We used Sec61 β as the ER marker, the first twenty residues of neuromodulin as the plasma membrane marker, and Tomm20 as the mitochondria marker.

Suppl. Figure 8: b is MyxV, not also HHV8 as said in the legend.

The error in the figure legend has been corrected. Please note that Supplementary Fig. 8 is now Supplementary Fig. 9 in the revised version of the manuscript.

What does a negative intramembrane interaction (Fig. S11) mean?

By “negative intramembrane interaction” we meant the absence of interaction. The manuscript has been revised accordingly for clarity, not only in Supplementary Fig. 12 (previously S11) (page 44) but also everywhere else where interactions had been characterized as “positive” or “negative”. We hope the new wording is more accurate.

References

1. Perdiguero, B. *et al.* Deletion of the viral anti-apoptotic gene F1L in the HIV/AIDS vaccine candidate MVA-C enhances immune responses against HIV-1 antigens. *PloS One* **7**, e48524 (2012).
2. Postigo, A. & Way, M. The vaccinia virus-encoded Bcl-2 homologues do not act as direct Bax inhibitors. *J. Virol.* **86**, 203–213 (2012).

3. Schanzenbach, C., Schmidt, F. C., Breckner, P., Teese, M. G. & Langosch, D. Identifying ionic interactions within a membrane using BLaTM, a genetic tool to measure homo- and heterotypic transmembrane helix-helix interactions. *Sci. Rep.* **7**, 1–11 (2017).
4. Ruan, W., Becker, V., Klingmüller, U. & Langosch, D. The Interface between Self-assembling Erythropoietin Receptor Transmembrane Segments Corresponds to a Membrane-spanning Leucine Zipper. *J. Biol. Chem.* **279**, 3273–3279 (2004).
5. Grau, B. *et al.* The role of hydrophobic matching on transmembrane helix packing in cells. *Cell Stress* **1**, 90–106 (2017).
6. Andreu-Fernández, V. *et al.* The C-terminal Domains of Apoptotic BH3-only Proteins Mediate Their Insertion into Distinct Biological Membranes. *J. Biol. Chem.* **291**, 25207–25216 (2016).

REVIEWERS' COMMENTS

Reviewer #1 (Remarks to the Author):

The authors have taken care of the comments of this reviewer. Importantly, they show now data with a virus mutant, that clarifies better the role of the Bcl2 analogs during viral infection. This is an interesting study in the fields of virology and cell biology.

Reviewer #2 (Remarks to the Author):

The authors have addressed all my concerns. The additional experiments performed further improved an already very nice paper.

Marc Kvansakul

Reviewer #3 (Remarks to the Author):

The authors have satisfied all my previous concerns in a satisfactory fashion.

Response to Reviewers

Reviewer #1:

The authors have taken care of the comments of this reviewer. Importantly, they show now data with a virus mutant, that clarifies better the role of the Bcl2 analogs during viral infection. This is an interesting study in the fields of virology and cell biology.

Reviewer #2:

The authors have addressed all my concerns. The additional experiments performed further improved an already very nice paper.

Reviewer #3:

The authors have satisfied all my previous concerns in a satisfactory fashion.

We thank the reviewers for their complements, helpful feedback, and suggestions throughout the review process. We sincerely believe that their input has improved the quality of the manuscript.

Sincerely,

Luis Martinez Gil, PhD

Assistant Professor

Dep. Biochemistry and Molecular Biology

University of Valencia

+34 963543296

luis.martinez-gil@uv.es